# A numerical approach applied to three-dimensional wave scattering problems subjected to obliquely propagating incident waves

Hao Lv [1,2¤] *

1 College of Safety and Environmental Engineering, Shandong University of Science and Technology, Qingdao, China, 2 Department of Civil and Airport Engineering, Nanjing University of Aeronautics and Astronautics, Nanjing, China

¤ Current address: College of Safety and Environmental Engineering, Shandong University of Science and Technology, Huang dao District, Qingdao, Shandong Province, P. R. China
* lvhao_happy@sina.com

**Citation:** Lv H (2024) A numerical approach applied to three-dimensional wave scattering problems subjected to obliquely propagating incident waves. PLoS ONE 19(6): e0304721. https://doi.org/10.1371/journal.pone.0304721

**Data Availability Statement:** All relevant data are within the manuscript and its S1 Data.

**Funding:** The authors gratefully acknowledge the financial support of this work by the Postdoctoral Fellowship Program of CPSF (No. GZC20231496) and the National Natural Science Foundation of

## Abstract

Accurately modeling artificial boundary conditions and wave inputs is paramount for numerical simulations of wave scattering in semi-infinite domains within seismic engineering. Traditionally, analysts focused on one- or two-dimensional free-field problems to determine wave inputs, primarily for vertically incident plane waves or obliquely incident waves parallel to two axes. However, these methods were inadequate for handling arbitrary incident directions in three-dimensional scenarios. This paper proposes a method for modeling seismic wave incidents in arbitrary directions. The basic theory of viscoelastic boundaries is leveraged, and a plane containing an arbitrary incident direction and the vertical coordinate axis is selected to establish a two-dimensional plane coordinate system. The two-dimensional free-field problem in this coordinate system is derived using the transfer matrix method. Subsequently, displacement, velocity, and stress are converted into the coordinate system where the three-dimensional calculation model is located, providing input for the three-dimensional scattering problem. Furthermore, the implementation of transmitting boundary conditions and viscoelastic boundary wave inputs is presented to enable incident wave scattering problems at any angle of the plane. The effect of oblique-incidence soil-structure dynamic interaction is also discussed, focusing on the parallel technology method adopted in this paper. With the relatively mature technology route and method, together with nuclear power systems and large-span deep-water bridge models, through examples of comparative analysis, qualitative and quantitative analyses are made on the impact on the soil mass, foundation, and structure when the seismic wave is an oblique incident.

## Introduction

Seismic monitoring data and experimental studies have provided compelling evidence that non-vertical incidence seismic waves exhibit more realistic seismic responses compared to

China (No. 52278359). The funders had no role in study design, data collection and analysis, or decision to publish, but they did have a role in the preparation and creation of the manuscript.

**Competing interests:** The authors have declared that no competing interests exist.

vertical incidence seismic waves. This can be attributed to the fault rupture mechanisms and the non-homogeneous nature of soil stratigraphy. Traditionally, it is assumed that seismic waves undergo multiple refractions as they traverse progressively softer soil layers, following Snell's law, which often results in vertical propagation. However, when seismic waves are incident at oblique angles, the varying angles of incidence cause reflection and refraction, leading to complex wave propagation patterns, including surface waves. These obliquely incident waves can have a significant impact on site response [1–4] as well as dynamic soil-structure interaction (SSI) behavior and ultimately the structural response [5–11].

In recent decades, significant advancements have been made in research pertaining to this topic. In the field of site response analysis, Trifunac [12] conducted an analytical investigation on the scattering of planar SH waves by semi-cylindrical and semi-elliptical canyons in an elastic homogeneous half-space. Luco and Barros [3] and Luco et al. [13] studied the scattering of cylindrical canyons in a laminar half-space under the influence of tilted incident P, SV, and SH waves, as well as valleys under the action of tilted incident P, SV, and SH waves. Subsequently, the same problem was addressed using the direct boundary element method (BEM) [14–16] and the indirect boundary element method (IBEM) [17]. Bielak et al. [18] employed local second-order impedance elements to simulate seismic SH wave propagation, considering oblique incidence. Karabalis and Beskos [19,20] utilized a combination of the time-domain boundary element method and the boundary element-finite element method to address the same problem for both rigid and flexible foundations. Huang et al. [21,22] analyzed the effect of incidence angle on the seismic performance of long-lined tunnels in an elastic half-space. Poursatip et al. [23,24] numerically investigated the influence of terrain irregularities on ground motion and local site response under obliquely incident P and SV waves in the frequency domain and elastic half-space. It is evident that most previous studies have either simplified the soil medium to a homogeneous elastic half-space in numerical simulations or addressed the problem in the frequency domain to account for stratification conditions. The difficulty lies in effectively integrating oblique incident waves into layered half-space models while applying suitable absorption boundary conditions in finite element analysis. As a result, many existing methods overlook the significance of soil stratigraphy. Additionally, soil non-linearities, which exert influence even under minimal strain, are often overlooked in frequency domain and boundary element analyses.

In practical engineering, there are three essential factors to consider in time-domain nonlinear site response and SSI analyses: (1) seismic motion input method, (2) absorbing boundary conditions, and (3) a suitable soil ontological model. When using numerical methods combined with artificial boundaries to simulate the infinite-domain plane-wave scattering problem, one of the key issues is how to provide fluctuation input to the artificial boundaries that is related to their characteristics. For transmissive boundaries, fluctuation input is achieved directly through free-field displacements, while for viscoelastic boundaries, it is achieved through equipotential forces, derived from the displacements, velocities, and stresses obtained from the free-field analysis [25]. Free-field analysis of transversely homogeneous horizontally layered media in the far and boundary zones presents a one-dimensional problem for vertically incident waves and a two-dimensional problem for obliquely incident waves. To solve this problem, Liu [26] proposed a simple spatially one-dimensional time-domain computational method by combining the concentrated mass finite element method and time-centered differencing. This method establishes a set of equations of motion for the nodes. According to Snell's law, the motion of a neighboring node in the horizontal direction can be expressed by the motion of the neighboring moment of that node, transforming the spatial two-dimensional system of equations for solving the node motion into a spatial one-dimensional system. Solving this system provides the motion of a vertical column of nodes in the free

wave field. Finally, according to the characteristics of traveling wave propagation, the free wave field of the entire space can be determined easily, transforming the spatial two-dimensional problem of calculating the horizontal layered half-space free-field during oblique seismic wave incidence into a simple spatial one-dimensional problem. Additionally, the free-field can be solved in the frequency domain using the transfer matrix method [27,28], and the free-field is obtained by the Fourier inverse transform, which offers higher accuracy than the numerical method in the time domain. Although it was commonly believed that the frequency domain solution of the two-dimensional free-field is less efficient than the numerical method in the time domain, this is not accurate. The motion of each point on the $z$- axis (a one-dimensional problem) can first be determined using the transfer matrix method, and then the free-field of the entire space can be extrapolated according to the apparent velocities of the $x$- and $y$-directions.

For the analysis of three-dimensional fluctuation scattering problems, the conventional approach was to use free-field analysis in one or two dimensions, typically considering perpendicular incidence or incidence parallel to one of the coordinate planes containing the vertical axis. However, this approach limits the incident directions of fluctuations in three-dimensional space and fails to accurately capture the true three-dimensional effects of ground vibration input. To address this limitation, it is necessary to analyze the free-field at arbitrary fluctuation incident directions in three-dimensional space, providing inputs for the analysis of fluctuation scattering in three-dimensional problems. Huang [29] proposed a solution for homogeneous half-space free-field incidents in any direction by combining the direct incident wave with the reflected wave from the ground surface. However, when dealing with the free-field analysis of horizontally layered sites, this method faces difficulties in tracking the complex reflected and transmitted waves, making it more challenging to solve the problem.

In this study, a two-dimensional planar coordinate system is defined by selecting a plane containing an incident direction and the longitudinal axis. The transfer matrix method is employed to solve the two-dimensional free-field problem within this coordinate system. Subsequently, the displacements, velocities, and stresses are converted to the coordinate system of the three-dimensional computational model, serving as inputs for the three-dimensional scattering analysis. Additionally, fluctuations from transmissive and viscoelastic boundaries are considered in the study.

## Theoretical approaches

In this study, the analytical solution of the 3D layered half-space free-field response [27,28,30] was utilized to investigate the impacts of obliquely incident P-waves and SV-waves. To accomplish this, the free-field response was integrated with an artificial boundary. A two-dimensional planar coordinate system was established, with a plane containing the incident direction and the longitudinal axis acting as the reference. Utilizing the transfer matrix method, the two-dimensional free-field within this coordinate system was determined. To furnish inputs for the 3D scattering problem, the displacement, velocity, and stress were converted into the coordinate system of the 3D computational model. This methodology facilitates the incorporation of diverse elements such as arbitrary soil non-homogeneities and nonlinearities, as well as structures like buildings, tunnels, piles, and bridges, generating varied inputs from transmissive and viscoelastic boundaries.

### Free-field calculations for seismic waves incident at arbitrary angles

As shown in Fig 1, the SSI model is an example; the coordinate system $oxyz$ was established, and the seismic wave input direction was described by $\theta_1$ and $\theta_2$. For the transmission artificial

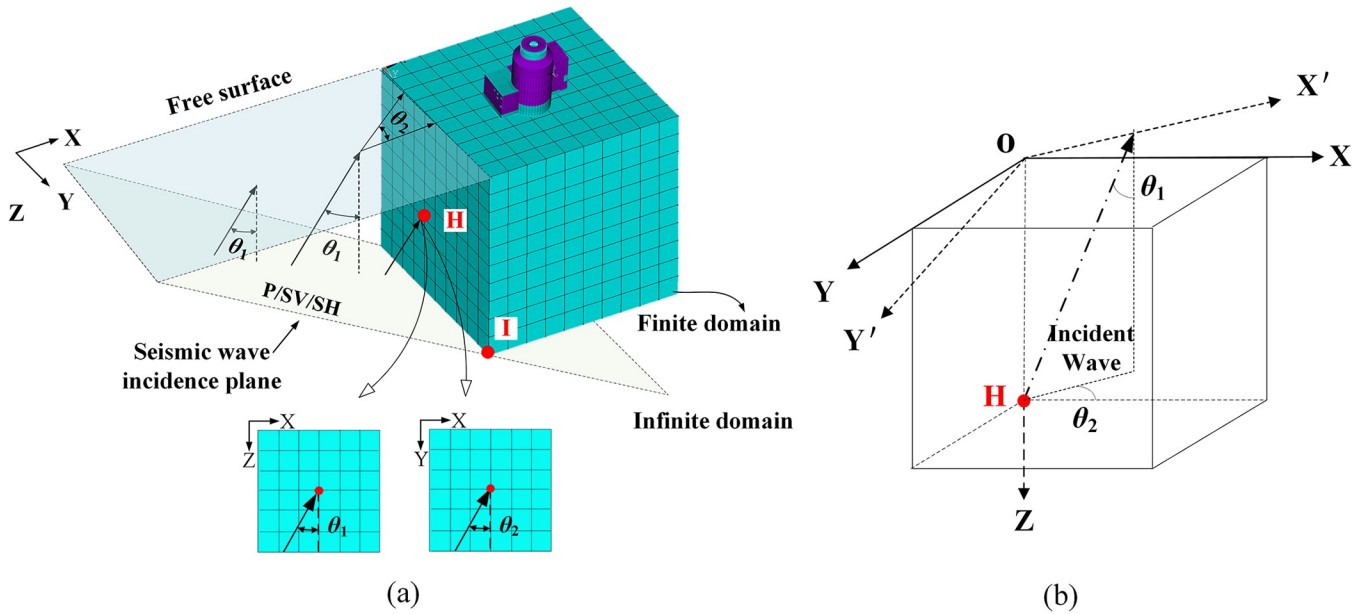

**Fig 1. Schematic diagram of soil structure interaction model under spatial incidence of seismic wave.** (a) The coordinate system oxyz is established, and the seismic wave input direction is described by θ1 and $\theta_2$. (b) Take the solution of the free field reference point H as an example, set its coordinates under the *xyz* coordinate system as and the coordinate system as the overall coordinate system.

boundary, the free-field displacement response at each boundary point needs to be calculated, and for the viscous-spring artificial boundary condition, the free-field displacement, velocity, and stress at each point on the boundary surface need to be obtained.

Using the free-field reference point **H** as a case study, designate its coordinates within the *xyz* coordinate system, with this coordinate system serving as the overarching reference frame. Proceed to construct the soil body model within the same *xyz* coordinate system, followed by discretization to determine the coordinates of individual nodes within the soil body. Rotate the *xyz* coordinate system around the z-axis counterclockwise by $\theta_2$ to become the new coordinate system *x′y′z′* (counterclockwise rotation is positive). The coordinates under the coordinate system *ox′y′z′* coordinate system is (*x′y′z′*), and the coordinate transformation relation is:

$$\left.\begin{array}{l} x' = x\cos\theta_2 - y\sin\theta_2 + I_y\sin\theta_2 \\ y' = x\sin\theta_2 + y\cos\theta_2 \\ z' = z \end{array}\right\} \tag{1}$$

where $I_y$ is the length of the model in the *y*-direction. Here, the $I_y\sin\theta_2$ translation term causes the wavefront to arrive at the corner point *I* below the left front of the soil body at time zero. In the *ox′y′z′* coordinate system, the transfer matrix method for horizontally layered media can be used to find the free-field response at point **H**, which yields the displacement $\mathbf{u}'_f$, velocity $\dot{\mathbf{u}}'_f$, and stress $\boldsymbol{\sigma}'_f$ at this point. Then, by coordinate system transformation, the displacement $\mathbf{u}'_f$, velocity $\dot{\mathbf{u}}'_f$ $\dot{\mathbf{u}}'_f$, and stress $\boldsymbol{\sigma}'_f$.

$$\begin{cases} u_x = u'_{x'}\cos\theta_2 + u'_{y'}\sin\theta_2 \\ u_y = -u'_{x'}\sin\theta_2 + u'_{y'}\cos\theta_2 \\ u_z = u'_{z'} \end{cases}, \begin{cases} \dot{u}_x = \dot{u}'_{x'}\cos\theta_2 + \dot{u}'_{y'}\sin\theta_2 \\ \dot{u}_y = -\dot{u}'_{x'}\sin\theta_2 + \dot{u}'_{y'}\cos\theta_2 \\ \dot{u}_z = \dot{u}'_{z'} \end{cases} \tag{2}$$

Here $\boldsymbol{\sigma}_f = \boldsymbol{\beta}\boldsymbol{\sigma}'_f\boldsymbol{\beta}^T$,

$$\boldsymbol{\beta} = \begin{bmatrix} \cos q_2 & \sin q_2 & \mathbf{0} \\ -\sin q_2 & \cos q_2 & \mathbf{0} \\ \mathbf{0} & \mathbf{0} & \mathbf{1} \end{bmatrix} \tag{3}$$

$$\boldsymbol{\sigma}_f = \begin{bmatrix} \sigma_{xx} & \tau_{xy} & \tau_{xz} \\ \tau_{yx} & \sigma_{yy} & \tau_{yz} \\ \tau_{zx} & \tau_{zy} & \sigma_{zz} \end{bmatrix}, \boldsymbol{\sigma}'_f = \begin{bmatrix} \sigma_{x'x'} & \tau_{x'y'} & \tau_{x'z'} \\ \tau_{y'x'} & \sigma_{y'y'} & \tau_{y'z'} \\ \tau_{z'x'} & \tau_{z'y'} & \sigma_{z'z'} \end{bmatrix} \tag{4}$$

### For simulating wave motion of the internal node

This section presents a brief overview of the Finite Element Method (FEM) used in a unified computational framework to simulate wave motion in horizontally layered dry soil (rock) sites. To improve computational efficiency, a Partitioned Analysis approach for Soil-Structure Interaction (SSI), known as PASSI, was employed. For a more detailed exploration of the theoretical aspects, readers are encouraged to refer to our earlier publications by references [31–33].

### Implementation of multi-transmitting artificial boundaries

Fig 2 illustrates the geometric relationship of the multi-transmitting boundary. This study assumes the soil in the boundary regions behaves linearly. Consequently, the governing equations for the boundary nodes can be expressed using the Multi-Transmitting Formula (MTF) proposed by Liao [34,35].

$$u^s((p+1)\Delta t, 0) = \sum_{j=1}^{N} (-1)^{j+1} C_j^N u^s((p+1-j)\Delta t, -jc_a\Delta t) \tag{5}$$

In Eq (5), $s$ superscript, represents the outgoing wave (or scattering wave). The displacements $u^s(t,x)$ of the outgoing wave (or scattering wave) can be expressed as a function of time: $t = p\Delta t$ and $x = -jc_a\Delta t$. Here, $\Delta t$ is the time step and $p$ is an integer. The coordinate $x = -jc_a\Delta t$

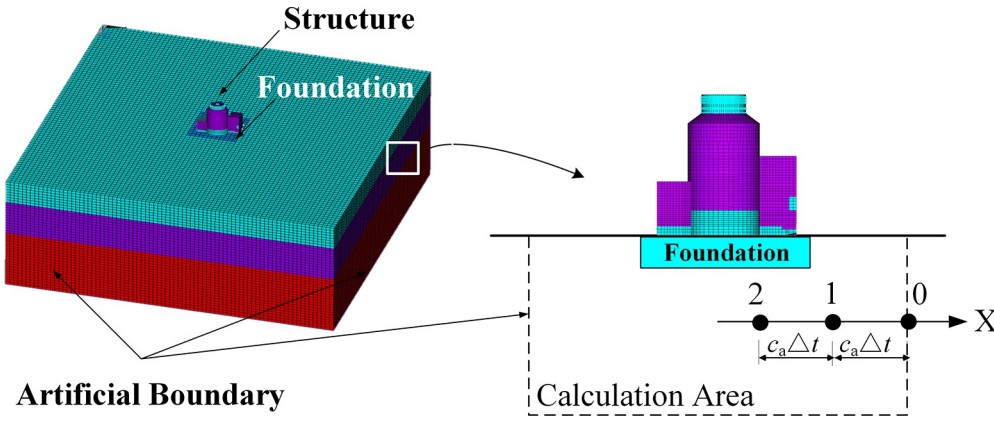

**Fig 2. Geometrical relation of the multi-transmitting boundary.**

represents the sampling points on the *x*-axis of a local coordinate that is perpendicular to the artificial boundary of the considered boundary point and points to the external infinite domain (as shown in Fig 2). $C_j^N$ is a binomial coefficient, $c_a$ is the artificial speed, and $N$ is the approximation order of the MTF. The most practical form of the local artificial boundary condition is a second-order MTF, which can be calculated via:

$$u^s((p+1)\Delta t, 0) = 2u^s(p\Delta t, -c_a\Delta t) - u^s((p-1)\Delta t, -2c_a\Delta t) \tag{6}$$

To implement Eq (6), an interpolation scheme is required to express $u^s(t, -jc_a\Delta t)$ in terms of $u(t, -n\Delta x)$. Here, we implement the calculation using the following interpolation. The displacements of the scattering wave at the nodal points can be expressed as:

$$u_n^{sp} = u^s(p\Delta t, -n\Delta x) \tag{7}$$

Using quadratic interpolation, the displacements at the computational points can be written as:

$$u^s(p\Delta t, -jc_a\Delta t) = \sum_{n=0}^{2j} t_{j,n} u_n^{sp} \quad (j = 1, 2) \tag{8}$$

Substituting Eq (8) into Eq (6), the second-order MTF can be written as

$$u_0^{s(p+1)} = \sum_{n=0}^{2} t_{1,n} u_n^{sp} + \sum_{n=0}^{4} t_{2,n} u_n^{s(p-1)} \tag{9}$$

For the external loads exerted on the structure or soil interior nodes, the total displacements are equal to displacements of the outgoing waves and can be calculated directly using Eq (9). For the scattering problem in which the input seismic wave impinges on the artificial boundary, wave-field decomposition was needed to obtain the total displacement using Eq (10). In this study, the total displacement of the boundary regions was decomposed into the free-field displacement and the scattering displacement in the side boundary region, which means that

$$u = u^s + u^f \tag{10}$$

where $u^s$ represents the scattering displacement and $u^f$ represents the free-field displacement and can be calculated by one-dimensional soil layer analysis when wave impinges vertically on the boundary.

## Implementation of viscous-spring artificial boundaries

The schematic diagram of the finite calculation area and the equivalent spring-damper system on the artificial boundary was illustrated in Fig 3. It is assumed that the soil in the boundary region can be modelled linearly. The values of the tangential and normal stiffness on the artificial boundary were determined using Eq (11):

$$\left.\begin{array}{ll} K_N = \alpha_N \dfrac{G}{R} & C_N = \rho c_P \\[2mm] K_T = \alpha_T \dfrac{G}{R} & C_T = \rho c_S \end{array}\right\} \tag{11}$$

where $R$ denotes the distance from the scattered wave source to the artificial boundary, $c_s$ is the shear wave's velocity of the medium and $c_p$ is the P wave velocity of the media. $G$ is the shear modulus and $\rho$ is the mass density of the medium. Here, $\alpha_T$ and $\alpha_N$ are modified coefficients in the tangential and normal directions, respectively. In this study, $\alpha_T = 0.67$, $\alpha_N = 1.33$ [36].

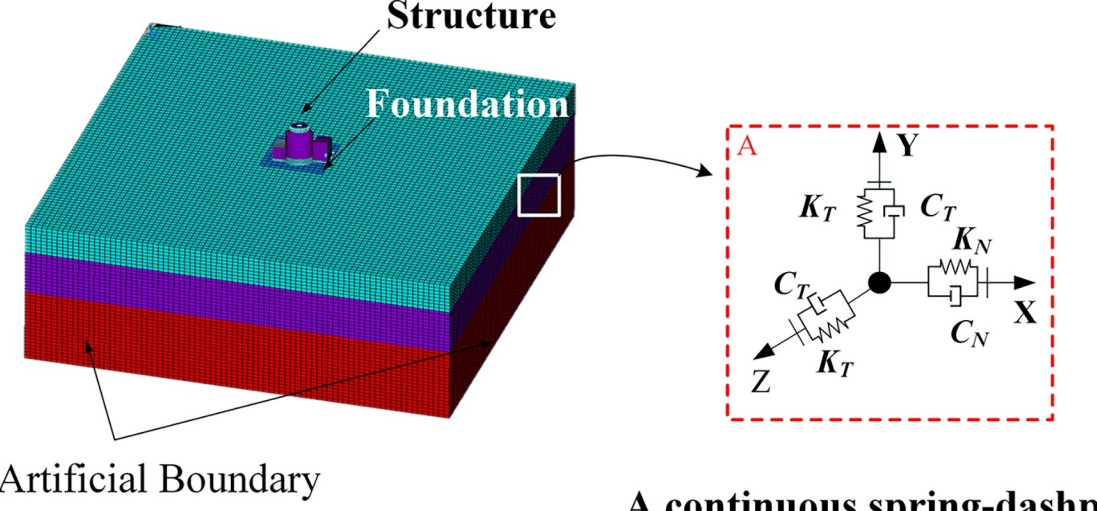

**Fig 3. Schematic diagram of the finite calculation area and equivalent spring-damper system on the artificial boundary.**

Seismic wave input can be realized by transforming into tangential and normal equivalent nodal forces:

$$\boldsymbol{\sigma}_i(p) = \boldsymbol{\sigma}_f(x, y, p) + c\dot{\mathbf{u}}_f(x, y, p) + k\mathbf{u}_f(x, y, p) \tag{12}$$

where $\sigma_i(p)$ denotes the equivalent load of point $i$ on the boundary at time $p$; $\sigma_f(x,y,p)$ is the stress in the continuum generated by the original free-field; $k$ is the spring constant and $c$ is the damping constant. $u_f(x,y,p)$ and $\dot{u}_f(x, y, p)$ represent displacement and velocity, respectively, corresponding to the free-field motion of point $i$.

The seismic motion at the artificial boundary was applied as equivalent forces to the boundary nodes. For vertical incident seismic excitation, the equivalent force $\mathbf{F}_B$ can be expressed as

$$\mathbf{F}_B = (\boldsymbol{\sigma}_f + k\mathbf{u}_f + c\dot{\mathbf{u}}_f)\mathbf{A} \tag{13}$$

where $\mathbf{A}$ represents the total area of all elements around the node considered on the artificial boundary. Eq (10) was used to calculate the time-history response of the boundary nodes. In this study, the spring and damping on the boundary should be considered in the stiffness matrix and damping matrix.

Fig 4 shows the direction of application of the equivalent force on the boundary surface, the stress components on the artificial boundary surface, and their positive directions [37]. According to the rules for determining the sign of the stress components, it is necessary to ensure that the displacement, velocity, and stress of the equivalent load applied on the corresponding artificial boundary are consistent with the original free-field. The method of applying the equivalent loads should be carried out as follows:

$$F_I = (\boldsymbol{\sigma}_f + k\mathbf{u}_f + c\dot{\mathbf{u}}_f)A \tag{14}$$

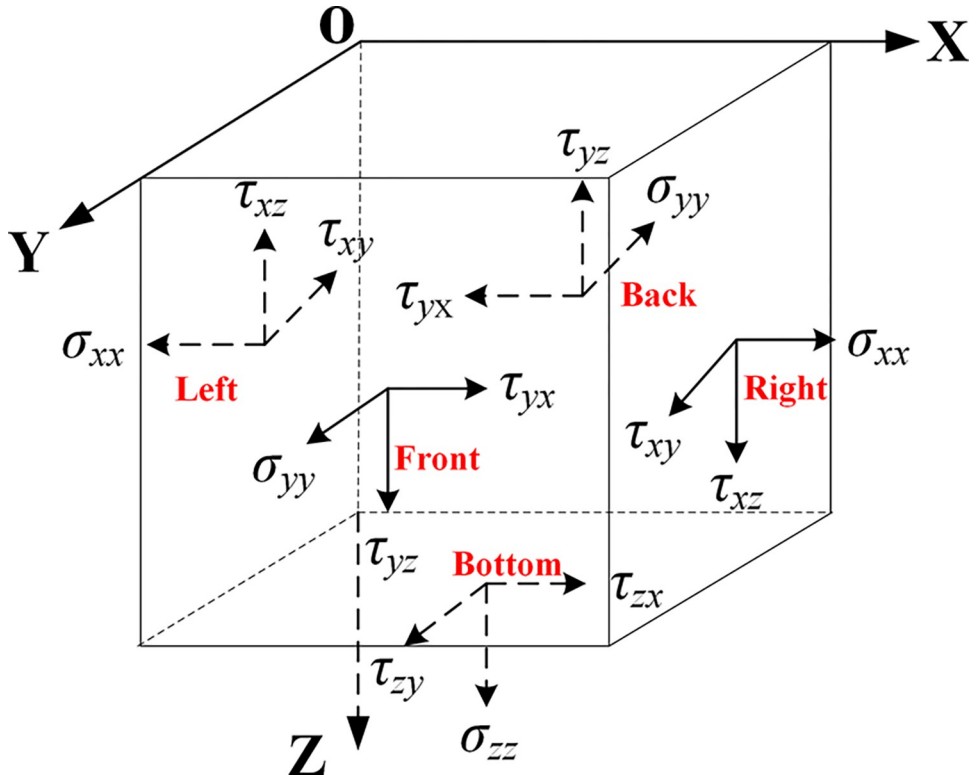

**Fig 4. Equal force and stress direction of interfaces in all directions.**

Left border:

$$
\left.\begin{array}{l}
F_{Ix} = (-\sigma_{xx} + k_N u_x + c_N \dot{u}_x)A \\
F_{Iy} = (-\sigma_{xy} + k_T u_y + c_T \dot{u}_y)A \\
F_{Iz} = (-\sigma_{xz} + k_T u_z + c_T \dot{u}_z)A
\end{array}\right\}
\tag{15}
$$

Back border:

$$
\left.\begin{array}{l}
F_{Ix} = (-\tau_{yx} + k_T u_x + c_T \dot{u}_x)A \\
F_{Iy} = (-\sigma_{yy} + k_N u_y + c_N \dot{u}_y)A \\
F_{Iz} = (-\sigma_{yz} + k_T u_z + c_T \dot{u}_z)A
\end{array}\right\}
\tag{16}
$$

Right border:

$$
\left.\begin{array}{l}
F_{Ix} = (\sigma_{xx} + k_N u_x + c_N \dot{u}_x)A \\
F_{Iy} = (\tau_{xy} + k_T u_y + c_T \dot{u}_y)A \\
F_{Iz} = (\tau_{xz} + k_T u_z + c_T \dot{u}_z)A
\end{array}\right\}
\tag{17}
$$

Front border:

$$
\left.
\begin{aligned}
F_{Ix} &= (\tau_{yx} + k_T u_x + c_T \dot{u}_x) A \\
F_{Iy} &= (\tau_{yy} + k_N u_y + c_N \dot{u}_y) A \\
F_{Iz} &= (\tau_{yz} + k_T u_z + c_T \dot{u}_z) A
\end{aligned}
\right\}
\tag{18}
$$

Bottom border:

$$
\left.
\begin{aligned}
F_{Ix} &= (\tau_{zx} + k_T u_x + c_T \dot{u}_x) A \\
F_{Iy} &= (\tau_{zy} + k_T u_y + c_T \dot{u}_y) A \\
F_{Iz} &= (\tau_{zz} + k_N u_z + c_N \dot{u}_z) A
\end{aligned}
\right\}
\tag{19}
$$

where $A$ represents the area represented by the boundary nodes. For the prongs and corners consider applying the stiffness $k$ on each face as well as the damping $c$ at the same time.

When the SH wave is incident at spatial angle, the free-field displacement $\mathbf{u}'_f$, velocity $\dot{\mathbf{u}}'_f$, and stress $\boldsymbol{\sigma}'_f$ in the $ox'y'z'$ coordinate system was obtained by the transfer matrix method for horizontal layered media. And the displacement, velocity and stress coordinates are transformed to the $oxyz$ coordinate system to obtain.

$$
\begin{cases}
u_x = u_{y'} \sin\theta_2 \\
u_y = u_{y'} \cos\theta_2 , \\
u_z = 0
\end{cases}
\quad
\begin{cases}
\dot{u}_x = \dot{u}'_{y'} \sin\theta_2 \\
\dot{u}_y = \dot{u}'_{y'} \cos\theta_2 \\
\dot{u}_z = 0
\end{cases}
\tag{20}
$$

$\boldsymbol{\sigma}_f = \boldsymbol{\beta} \boldsymbol{\sigma}_f \boldsymbol{\beta}^{\mathrm{T}}$ and then

$$
\sigma_f =
\begin{bmatrix}
\sigma_{xx} & \tau_{xy} & \tau_{xz} \\
\tau_{yx} & \sigma_{yy} & \tau_{yz} \\
\tau_{zx} & \tau_{zy} & \sigma_{zz}
\end{bmatrix}
=
\begin{bmatrix}
2\sin\theta_2\cos\theta_2\tau_{x'y'} & (\cos^2\theta_2 - \sin^2\theta_2)\tau_{x'y'} & \sin\theta_2\tau_{y'z'} \\
(\cos^2\theta_2 - \sin^2\theta_2)\tau_{x'y'} & -2\sin\theta_2\cos\theta_2\tau_{x'y'} & \cos\theta_2\tau_{y'z'} \\
\sin\theta_2\tau_{y'z'} & \cos\theta_2\tau_{y'z'} & 0
\end{bmatrix}
\tag{21}
$$

Here,

$$
\boldsymbol{\sigma}'_f =
\begin{bmatrix}
0 & \tau_{x'y'} & 0 \\
\tau_{y'x'} & 0 & \tau_{y'z'} \\
0 & \tau_{z'y'} & 0
\end{bmatrix}
\tag{22}
$$

The boundary force in the case of spatial incidence of SH wave can be obtained from Eqs (14)–(22).

For the case of P/SV wave, the free-field in the coordinate system $ox'y'z'$, displacement $u'_{x'}$, $u'_{z'}$, velocity $\dot{u}'_{x'}$, $\dot{u}'_{z'}$, and stress $\boldsymbol{\sigma}'_f$ can also be obtained from the displacement, velocity and

stress in the coordinate system by the transformation of the coordinate system *oxyz*:

$$\begin{cases} u_x = u'_{x'}\cos\theta_2 \\ u_y = -u'_{x'}\cos\theta_2\,, \\ u_z = u'_{z'} \end{cases} \begin{cases} \dot{u}_x = \dot{u}'_{x'}\cos\theta_2 \\ \dot{u}_y = -\dot{u}'_{x'}\sin\theta_2 \\ \dot{u}_z = \dot{u}'_{z'} \end{cases} \tag{23}$$

$$\sigma_f = \begin{bmatrix} \sigma_{xx} & \tau_{xy} & \tau_{xz} \\ \tau_{yx} & \sigma_{yy} & \tau_{yz} \\ \tau_{zx} & \tau_{zy} & \sigma_{zz} \end{bmatrix} = \begin{bmatrix} \cos\theta_2\tau_{x'x'} & -\sin\theta_2\cos\theta_2\sigma_{x'x'} & \cos\theta_2\tau_{z'x'} \\ -\sin\theta_2\cos\theta_2\sigma_{x'x'} & \sin\theta_2\sigma_{x'x'} & -\sin\theta_2\tau_{z'x'} \\ \cos\theta_2\tau_{z'x'} & -\sin\theta_2\tau_{z'x'} & \sigma_{z'z'} \end{bmatrix} \tag{24}$$

## Algorithm validation

To validate the accuracy of the equations and the feasibility of the modelling calculations, simulations were carried out using ANSYS software [38], with reference to the examples provided by Huang et al. [39] for verification.

A homogeneous half-space finite element model is depicted in Fig 5. The soil properties used in the model are as follows: a mass density of 2630 kg/m$^3$, an elastic modulus of 32.5 GPa, and a Poisson's ratio of 0.22 [39]. The dimensions of the soil were 60m × 60m × 60m, and those of the soil were discretized into hexahedral solid elements with dimensions of 2m × 2m × 2m along the *x*-, *y*-, and *z*- directions [40]. An incident SV wave at an angle of 30° was simulated using the method derived in the preceding section 2. The resulting time history of vibration displacement caused by the incident wave is depicted in Fig 6.

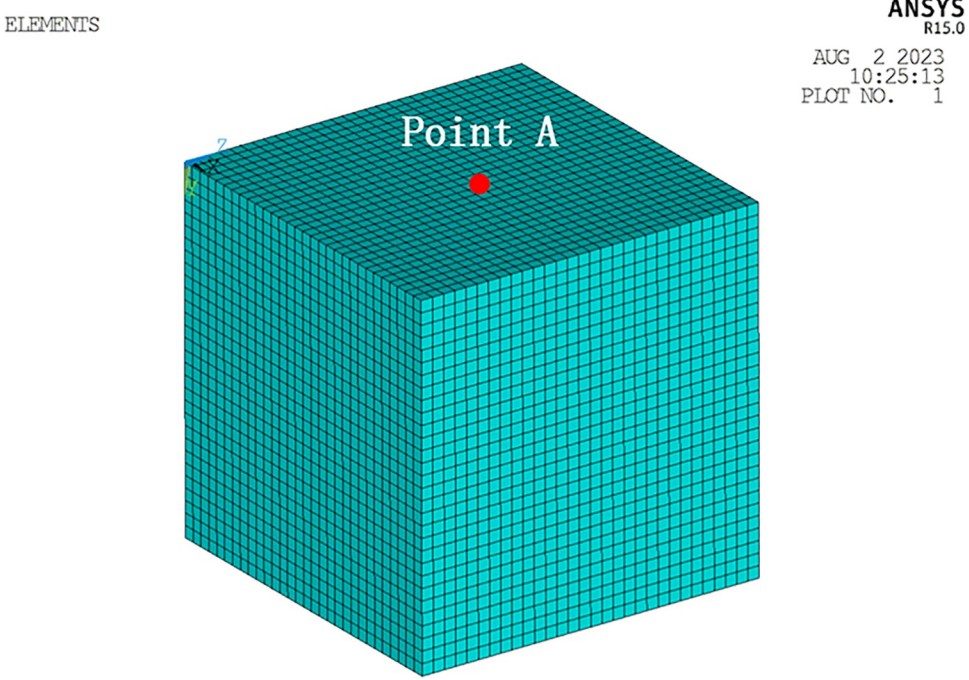

**Fig 5. Finite element model of the site.**

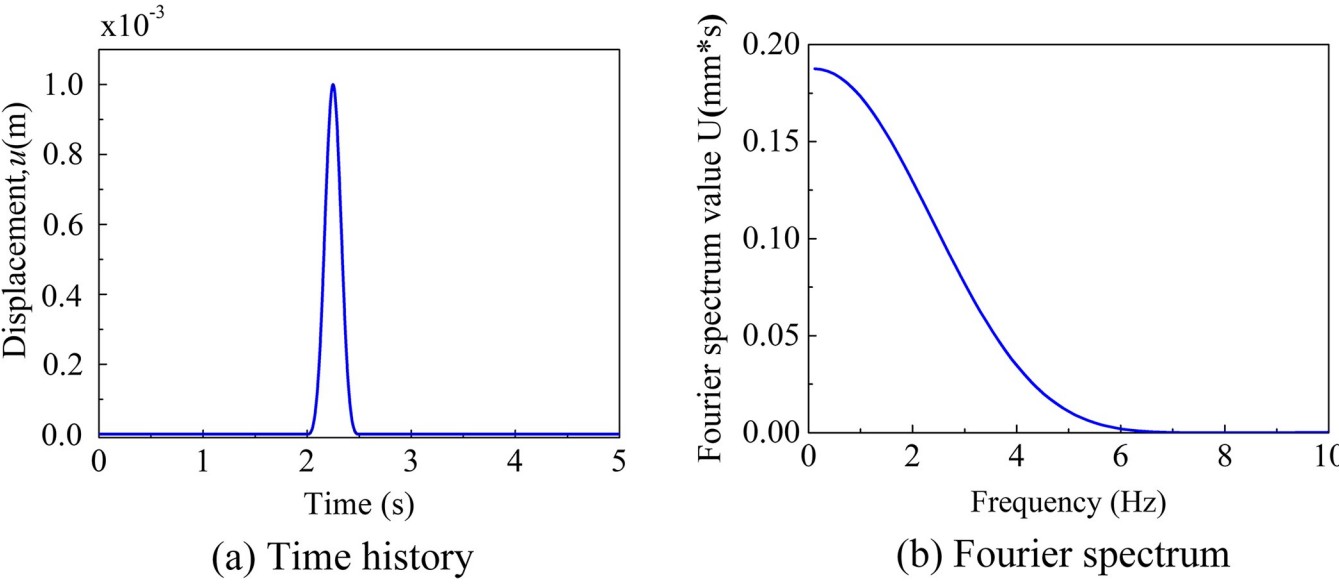

**Fig 6. Input wave.** (a) Time history of the incident wave. (b) The fourier spectrum of the incident wave.

Fig 7 presents a visually obscured depiction of the half-space displacement field of the SV wave under oblique incidence. The wavefronts of the incident SV wave, reflected SV wave, and reflected P wave are clearly discernible. Additionally, the horizontal displacement time course of reference point A is illustrated in Fig 8.

A comparison between the numerical and analytical solutions of the displacement response time series at the observation point reveals a strong agreement. The configuration of the viscous-spring artificial boundary and the computation of the equivalent nodal load effectively facilitate the calculation of obliquely incident seismic waves, demonstrating the high simulation accuracy of the method proposed in this paper.

The objective of this study was to develop analytical solutions for the free-field response of a three-dimensional (3D) layered half-space when subjected to obliquely incident P- and SV-

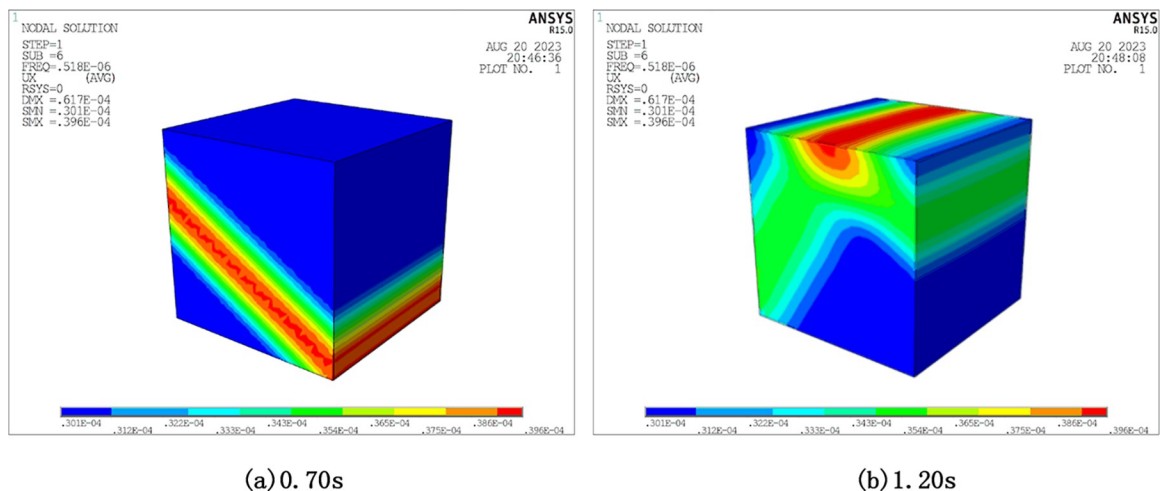

**Fig 7. Displacement of half space subjected to SV wave oblique incidence.** (a) At 0.70s. (b) At 1.20s.

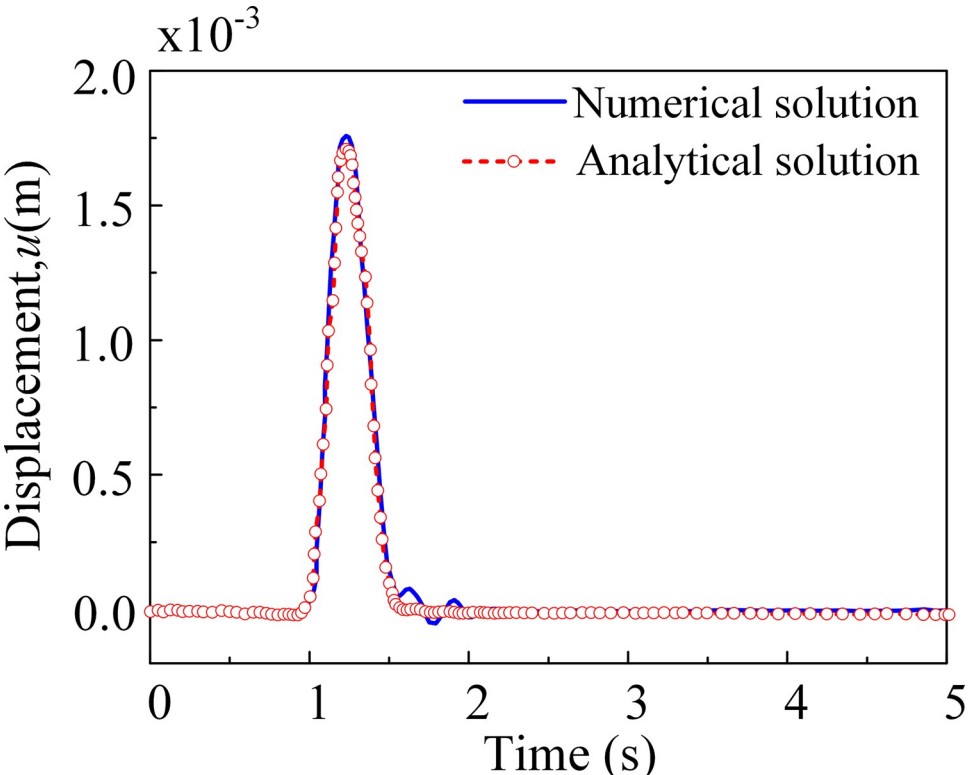

**Fig 8. The horizontal displacement time history of reference point A under SV wave oblique incidence.**

waves. These analytical solutions can then be incorporated into the lumped-mass explicit Finite Element Method (FEM) to propagate the incident waves to a specific region of interest. This region may involve various soil inhomogeneities, nonlinearities and include structures such as buildings, tunnels, piles, bridges, and so on, which act as scatterers.

## Analytical solutions for wave propagation in laminar elastic half-spaces

Wave potential theory, which has been extensively studied by Haskell [28], Aki and Richards [41], Achenbach [42], and Brekhovskikh [43], was used as an approach for calculating the phase and velocity of elastomers and surface waves in multilayer solid media in the frequency domain. Time-domain solutions can be obtained using the Fourier inverse transform. Zhao et al. [44] have demonstrated the efficacy of this method in various domains.

The following section presents a simple verification using an arithmetic example, and discusses the impact of the incident angle on the response. The boundaries were characterized as transmissive and viscoelastic, as previously described. The internal nodes are modeled using concentrated mass explicit finite elements. The analysis of wave scattering in different directions was conducted using a Fortran program (subroutines in the self-programmed software PASSI).

As illustrated in Fig 9, the 3D soil model has dimensions of 60 m × 60 m × 20 m, consisting of 72,000 elements and 78,141 nodes. The site soil has a mass density of 1600 kg/m$^3$, a shear wave velocity of 200 m/s, a compression wave velocity of 374 m/s, and a Poisson's ratio of 0.30.

The seismic excitation was represented by an impulse time function, as illustrated in Fig 10. The pulse width was 0.2 seconds, and the time step was $\Delta t = 2 \times 10^{-4}$s, with a total calculation time step of $n = 8192$.The cut-off frequency of 18 Hz was determined based on the material

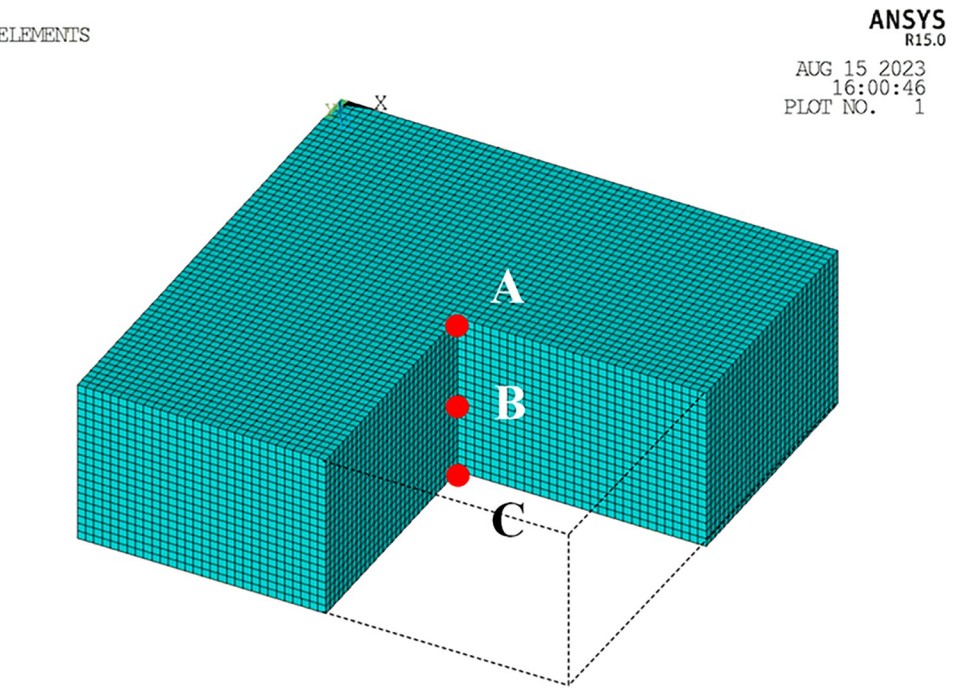

**Fig 9. An example model schematic diagram.**

properties of the site, and the grid spacing and time step can be directly determined using the reference [40].

The soil domain was discretized into hexahedral solid elements measuring 1.0 m × 1.0 m × 1.0 m in the $x$-, $y$-, and $z$-directions, ensuring accurate simulation of fluctuations with $\Delta x \leq V_s/(10 \times f_{max}) = 1.11$m (where $f_{max}$ represents the cut-off frequency and $V_s$ denotes the smallest shear wave velocity of the soil). For the soil, an explicit central difference format with a

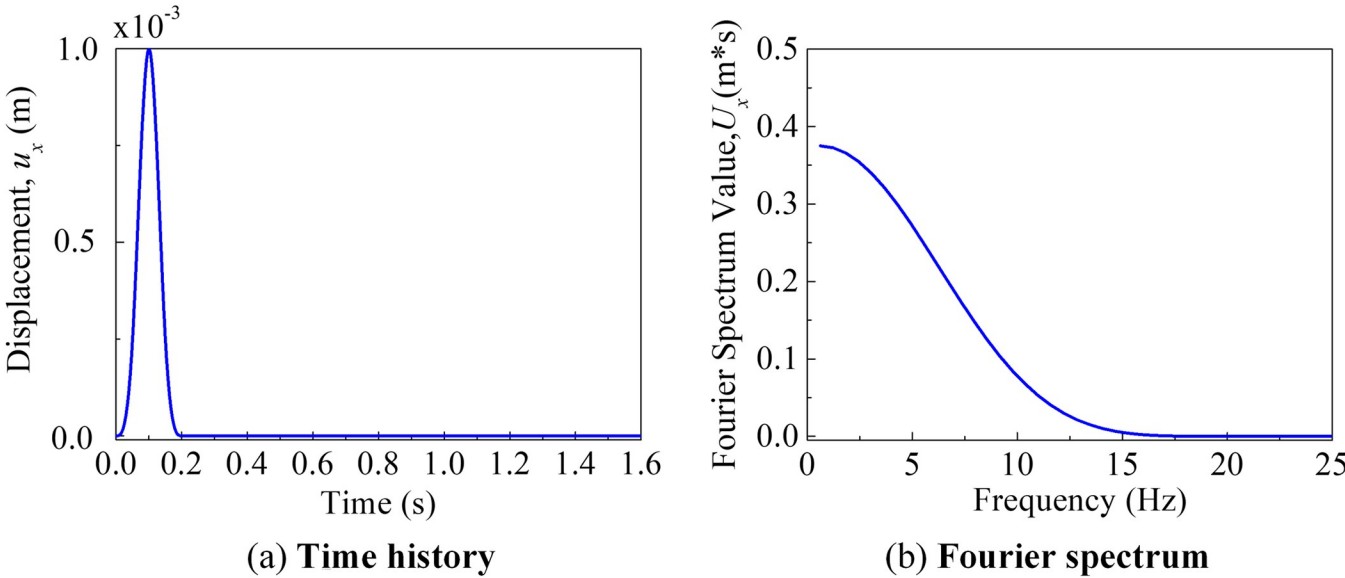

(a) **Time history**                    (b) **Fourier spectrum**

**Fig 10. Input plus wave.** (a) Time history of the plus wave. (b) The fourier spectrum of the plus wave.

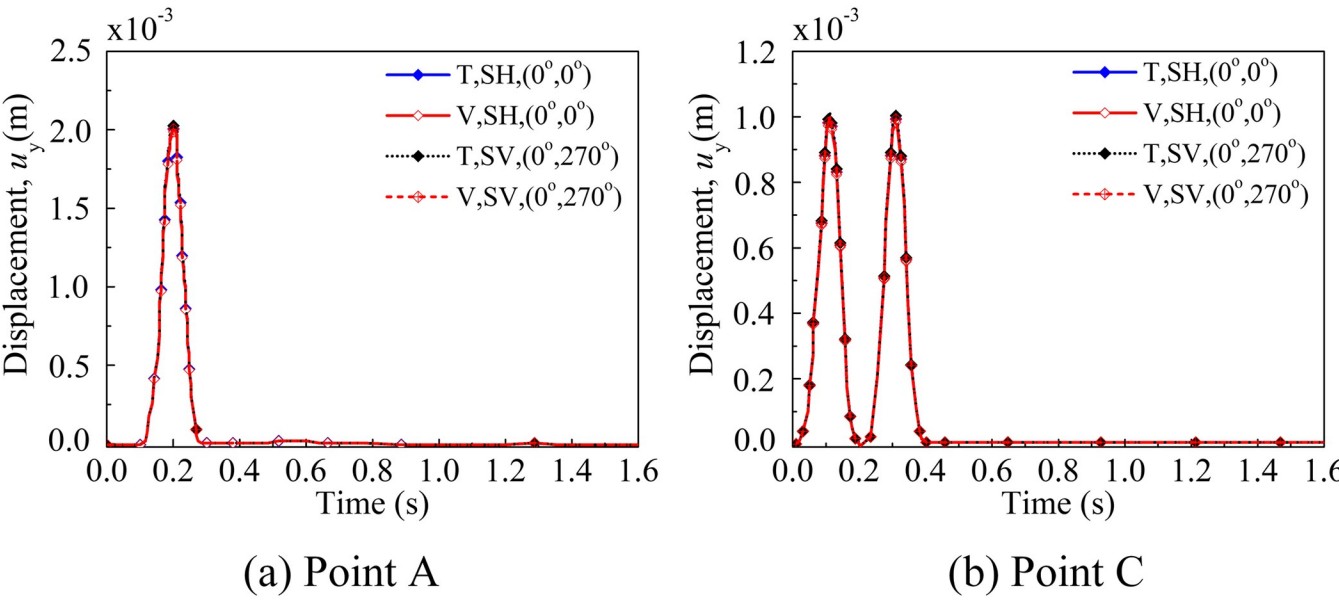

**Fig 11. Displacement in *y*-direction at observation points. Note: T and V represent the transmitting boundary and the viscous-spring artificial boundary.** (a) Displacement in *y*- direction at point A under the transmitting boundary and viscous-spring artificial boundary. (b) Displacement in *y*-direction at point C under the transmitting boundary and viscous-spring artificial boundary.

time step of $\Delta t = 2 \times 10^{-4}$s was adopted. The points A (30,30,0), B (30,30,10), and C (30,30,20) are selected for studying soil behavior as depicted in Fig 9. The viscous-spring artificial boundary conditions were employed at the site boundaries, with modified coefficients $\alpha_T$ and $\alpha_N$ representing the tangential and normal directions, respectively. In this study, $\alpha_T = 0.67$ and $\alpha_N = 1.33$.

The calculations presented in this study were conducted on a Dell Optiplex-3050 workstation equipped with an Intel Core i7 CPU operating at 3.40 GHz and 64 GB of RAM. The workstation runs Ubuntu 16.04 LTS as the operating system.

## Vertical incidence of SH and SV waves

According to the theory, the response of the SH wave ($\theta_1 = 0°$, $\theta_2 = 0°$) and SV wave ($\theta_1 = 0°$, $\theta_2 = 270°$) in the case of perpendicular incidence is identical, resulting in displacement solely in the *y*- direction. Utilizing the transmitting boundary and viscous-spring artificial boundary, the findings showcased in Fig 11 (T and V represent the transmitting boundary and the viscous-spring artificial boundary) were derived through the approach introduced in this paper. These outcomes exhibit alignment with the theoretical solution, thereby offering initial validation of the accuracy and reliability of the methodology and procedures delineated in this research.

Fig 12 depicts the displacement time histories of the reference points in the *x*-and *y*-directions for the SH wave incidence case, considering three spatial angle cases ($\theta_1 = 0°$, $\theta_2 = 30°$), and ($\theta_1 = 30°$, $\theta_2 = 0°$), respectively. When $\theta_2 = 0°$, the displacement in the *x*-direction was zero. Conversely, when $\theta_2 = 30°$, the displacement in the *x*-direction is 1.00 and the displacement in the *y*-direction is 1.732, aligning with the theoretical value of Eq (20) ($u_x = u'_{y'}\sin\theta_2 = 2\sin(30°) = 1.00$, $u_y = u'_{y'}\cos\theta_2 = 2\cos(30°) = 1.73$). The values obtained at the remaining points also align with the theoretical values, further validating the method proposed in this paper. As observed from the figure, $u_x$ and $u_y$ were solely related to $\theta_2$.

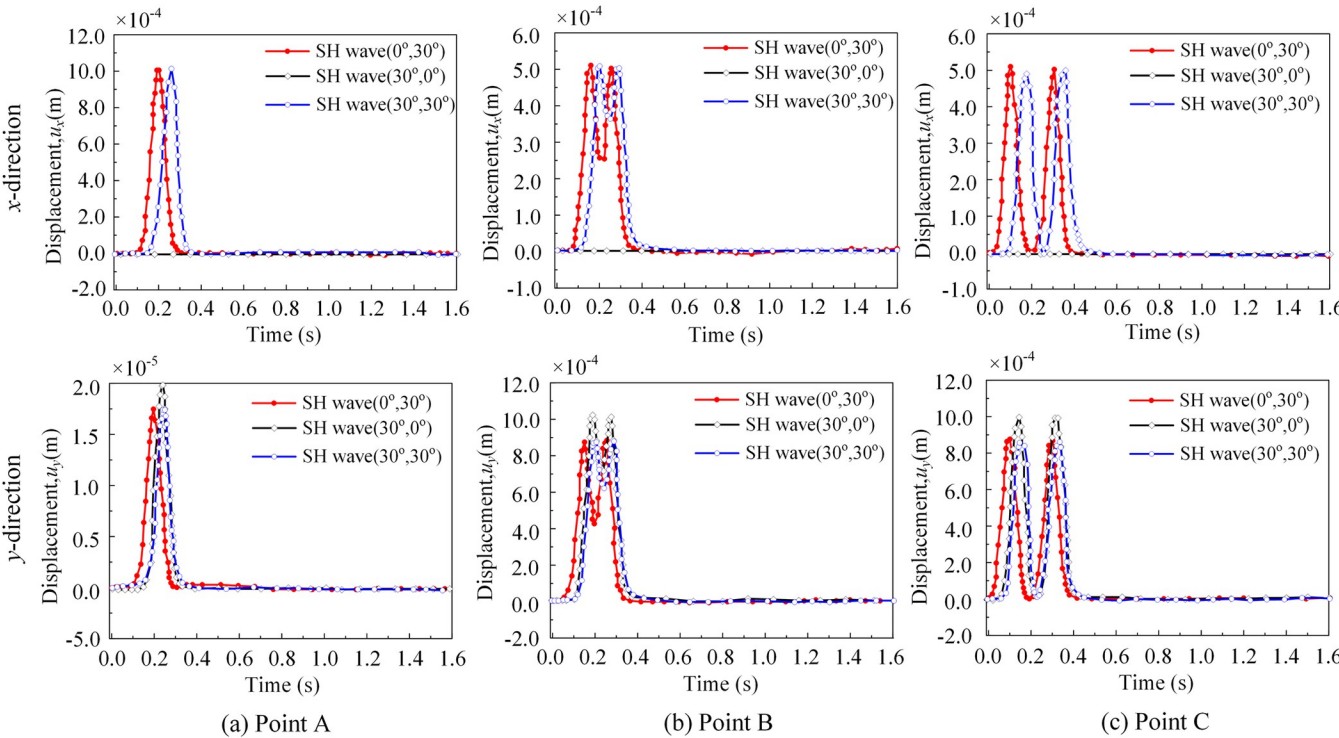

**Fig 12. Displacement of the observation points (SH wave).** (a) The displacement time histories of the point A in the *x*- and *y*-directions for the SH wave incidence. (b) The displacement time histories of the point B in the *x*- and *y*-directions for the SH wave incidence. (c) The displacement time histories of the point C in the *x*- and *y*-directions for the SH wave incidence.

Fig 13 illustrates the displacement time histories in three directions (*x*, *y*, and *z*) at the observation point, considering three spatial angle cases ($\theta_1 = 0°$, $\theta_2 = 20°$), ($\theta_1 = 20°$, $\theta_2 = 0°$), ($\theta_1 = 20°$, $\theta_2 = 20°$) for the SV wave incidence case. The surface reference point A's displacement corresponds to the theoretical value ($u_x = u'_{y'}\sin\theta_2 = 2\sin(20^o) = 0.68$, $u_y = u'_{y'}\cos\theta_2 = 2\cos(20^o) = 1.88$). From Fig 13, in the *y*-direction, unlike the case where only the incident direction parallel to the *xoz* coordinate plane was considered ($\theta_2 = 0°$), the displacement in the *y*-direction occurs even for SV wave incidence at $\theta_2 \neq 0°$. The values of $u_x$ and $u_y$ were related to both angles $\theta_1$ and $\theta_2$, whereas the value of $u_z$ was related to $\theta_1$ only.

## Example analysis

### Seismic response analysis of nuclear power structures

**Model and parameters.**   The finite element model of nuclear island buildings was established by ANSYS software. Nuclear island structures, characterized by their large size, large mass, and complex structure, have higher requirements for seismic performance. A three-dimensional finite element model of a nuclear power plant (NPP) was depicted in Fig 14. The total height of the structure was about 86.50 m, and the maximum plane size was 42.32 m × 90.80 m. The model consists of 194,802 degrees of freedom with 32,467 nodes and 35,162 elements (including mass elements). The material parameters were listed in Table 1.

The input seismic motion is defined at the bedrock, which was located at a depth of 60m and propagates vertically to calculate the free-field motion. The shear wave velocity at the bedrock (depth > 60m) is 8000 ft/s (2438.4 m/s). The total duration of the seismic motion was 1.2

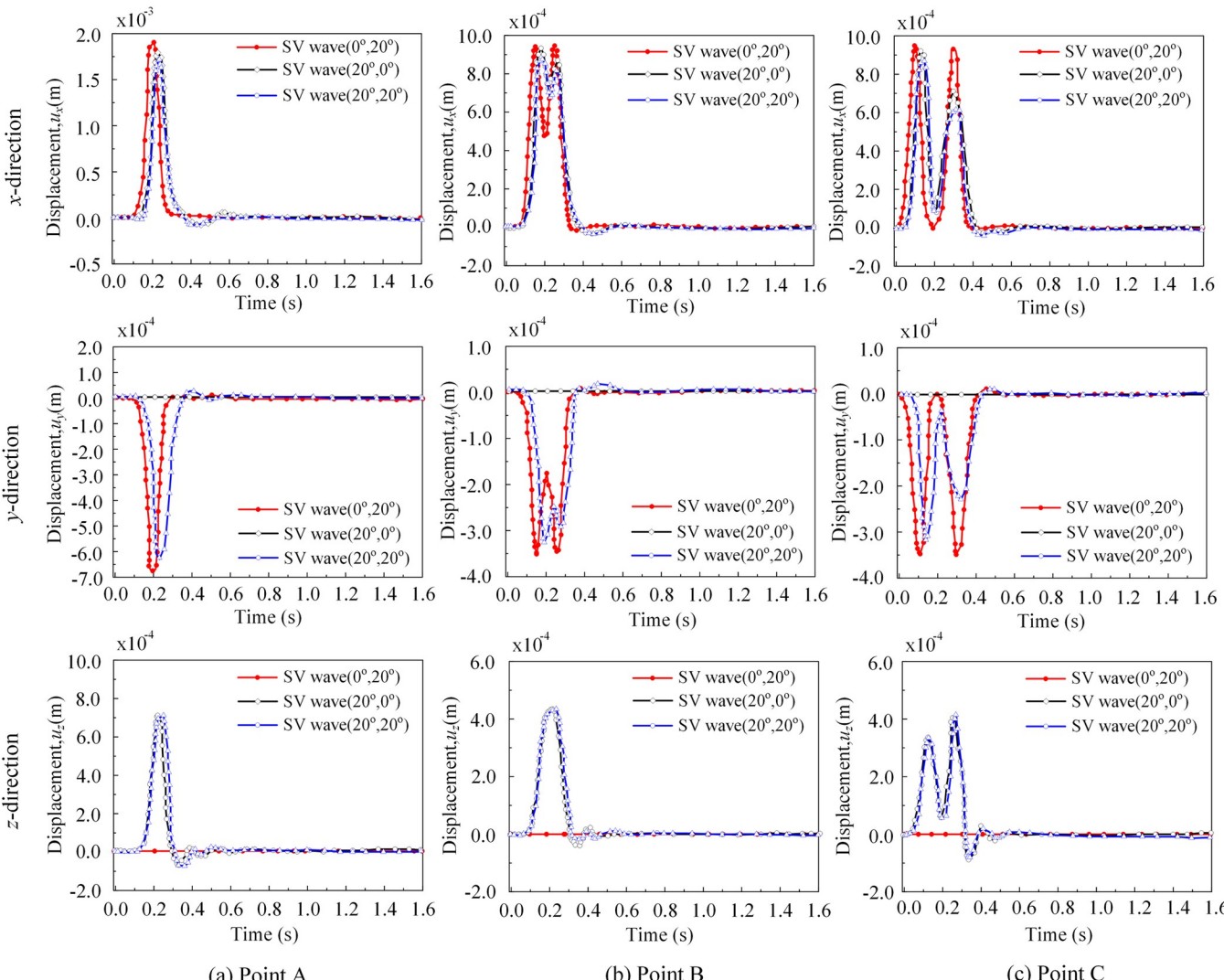

**Fig 13. Displacement of the observation points.** (a) The displacement time histories in three directions ($x$, $y$, and $z$) at the point A, considering three spatial angle cases ($\theta_1 = 0°$, $\theta_2 = 20°$), ($\theta_1 = 20°$, $\theta_2 = 0°$), ($\theta_1 = 20°$, $\theta_2 = 20°$) for the SV wave incidence. (b) The displacement time histories in three directions ($x$, $y$, and $z$) at the point B, considering three spatial angle cases ($\theta_1 = 0°$, $\theta_2 = 20°$), ($\theta_1 = 20°$, $\theta_2 = 0°$), ($\theta_1 = 20°$, $\theta_2 = 20°$) for the SV wave incidence. (c) The displacement time histories in three directions ($x$, $y$, and $z$) at the point C, considering three spatial angle cases ($\theta_1 = 0°$, $\theta_2 = 20°$), ($\theta_1 = 20°$, $\theta_2 = 0°$), ($\theta_1 = 20°$, $\theta_2 = 20°$) for the SV wave incidence.

$s$ ($\Delta t = 5 \times 10^{-5}$ $s$), and the cut-off frequency was approximately determined, as shown in Fig 15.

The size of the 3D soil model was set to 700m × 400m × 60m (with a rigid base size of 96m × 60m × 16m), consisting of 2,100,000 elements and 2,187,081 nodes. The soil profiles, shear wave velocity profiles, and relevant governing parameters were presented in Table 2 to ensure an accurate representation of the seismic wave propagation from the bottom soil layer to the surface. The maximum dimension $\Delta l$ of the mesh in the model was chosen to be 1/8 to 1/10 of the wavelength corresponding to the cut-off frequency. To model the soil (foundation) soil, a lumped-mass explicit Finite Element Method (FEM) was employed, utilizing an explicit central difference format with a time step $\Delta t$, where $\Delta t \leq \Delta l / c_{\max}$. Here, $c_{\max}$ represents the maximum shear wave velocity of the soil layers. Based on the soil properties provided in

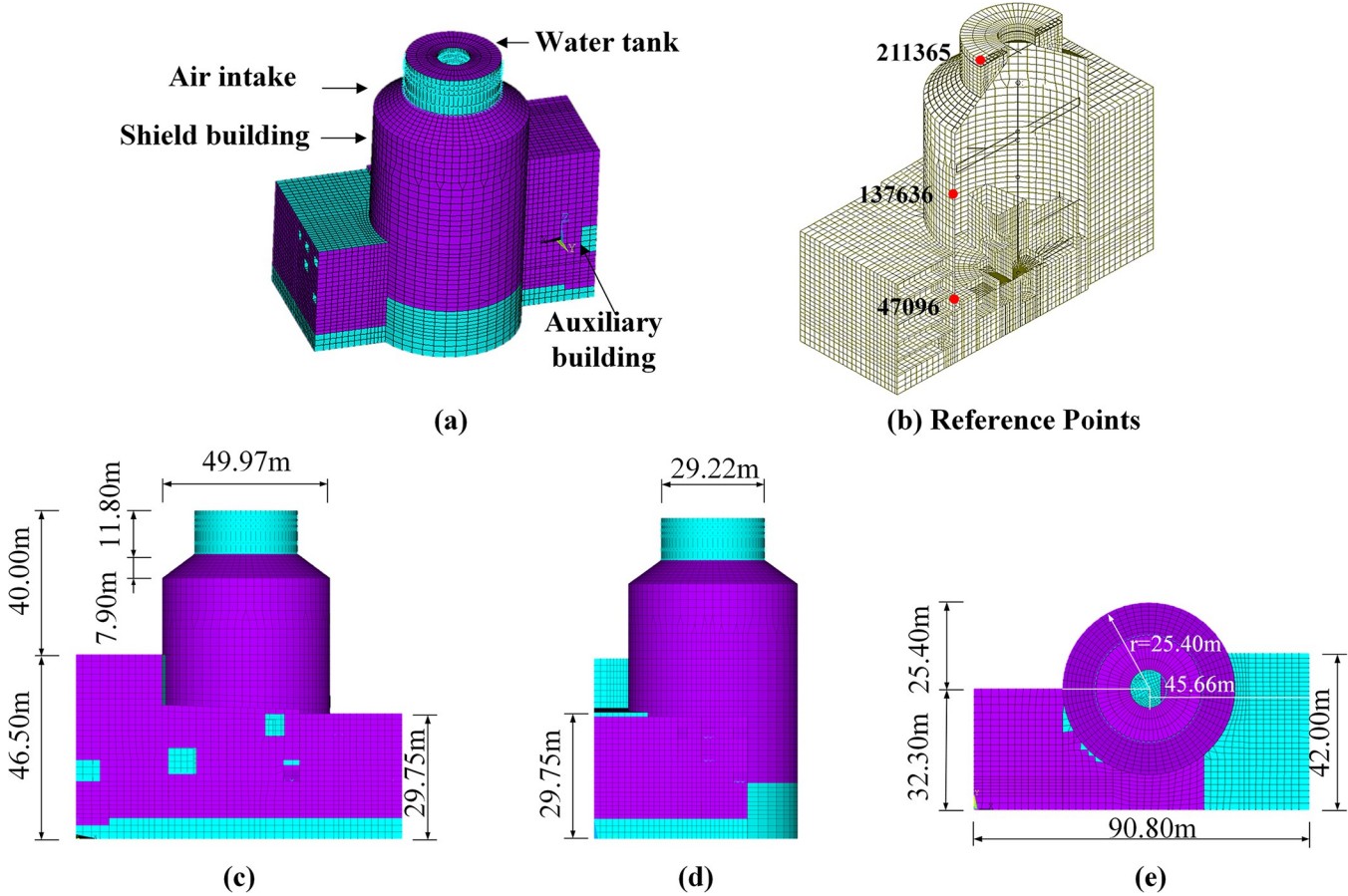

**Fig 14. An example model schematic diagram of the NPP.** (a) The three-dimensional finite element model of a nuclear power plant (NPP). (b) Structural sections and reference points. (c) Structure front elevation and dimensions. (d) Structural side elevations and dimensions. (e) Structure top view and dimensions.

Table 2 and considering the maximum frequency of the analysis, a soil element size of 2.0 m was selected. Consequently, the soil was discretized into 2.0 m × 2.0 m × 2.0 m hexahedral solid elements along the $x$-, $y$-, and $z$- directions.

All computational analyses in this study were conducted using the intrafield parallel computation method in the soil subsystem and the interfield parallel computation method between the soil and structure, as proposed in our previous work utilizing the PASS algorithm [31–33]. Along the $x$- axis direction, the soil subsystem was partitioned into three separate subdomains, named from left to right as soil subdomain 1, soil subdomain 2, and soil subdomain 3, as

**Table 1. Material parameters.**

| Material | Component | Elastic Modulus (N/m$^2$) | Poisson Ratio | Density (kg/m$^3$) |
|---|---|---|---|---|
| Concrete | Water Tank | $3.10×10^{10}$ | 0.17 | 2450 |
| | Cone Roof | $3.15×10^{10}$ | 0.17 | 2369 |
| | External cooling air inlet | $3.25×10^{10}$ | 0.17 | 2180 |
| | Concrete Containment Vessel | $3.30×10^{10}$ | 0.15 | 2168 |
| | Auxiliary Building | $2.50×10^{10}$ | 0.15 | 2400 |
| Steel | Steel Containment Vessel | $2.50×10^{10}$ | 0.30 | 7750 |

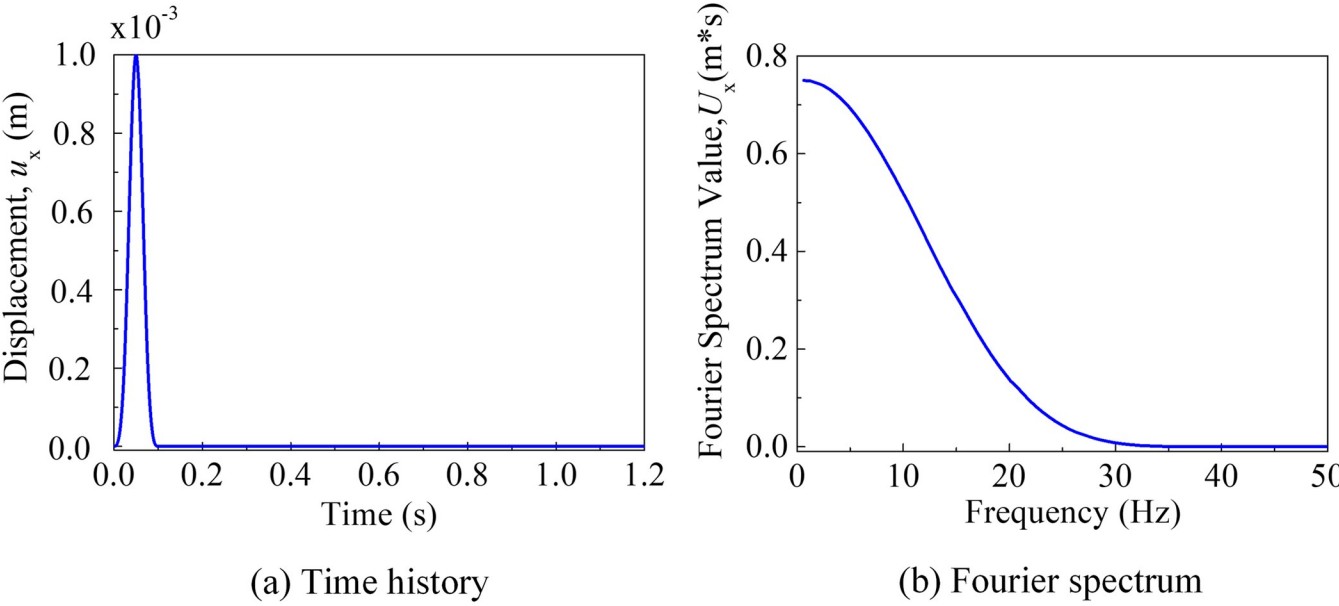

(a) Time history                    (b) Fourier spectrum

**Fig 15. The input wave and the cut-off frequency.** (a) Time history of the input wave. (b) The fourier spectrum of the input wave.

shown in Fig 16. The corresponding dimensions of each subdomain were 252m × 400m × 60m, 200m × 400m × 60m, and 252m × 400m × 60m, respectively. Overlapping layer elements were employed between adjacent subdomains. Each soil subdomain was associated with a distinct process. The structure, foundation, and soil subdomain 2 were all handled within the same process, while messages are passed between adjacent soil subdomains through the Message Passing Interface (MPI) protocol. For the soil, we employed an explicit central difference format with a time step of $\Delta t_1 = \Delta t = 5 \times 10^{-5}$ s. For the structure, we used a Newmark implicit integration format with a time step of $\Delta t_2 = 25\Delta t_1$, where $\Delta t_2 = 1.25 \times 10^{-3}$ s.

The number and coordinates of the reference points selected for the soil in this calculation example are depicted in Fig 16. In soil Subdomain 1, the reference points are assigned as follows: A11 (126,200,0), A12 (126,200,30), and A13 (126,200,60). In soil Subdomain 2, the reference points are A21 (428,200,0), A22 (352,200,16) (the reference point represents the displacement at the bottom of the assumed rigid foundation in this study), and A23 (352,200,60). Lastly, in soil Subdomain 3, the reference points are A31 (578,200,0), A32 (578,200,30), and A33 (578,200,60). Additionally, three observation points were selected from the upper, middle, and lower positions of the structure, with node numbers 211365, 137636, and 47096, respectively, as depicted in Fig 14(B).

**Table 2. Parameters of soil.**

| Layer | Thickness (m) | Density (kg/m³) | Shear wave velocity $V_s$(m/s) | Compressional wave velocity $V_p$ (m/s) | Poisson ratio | Damping factor |
|---|---|---|---|---|---|---|
| 1 | 20 | 1900.0 | 330 | 2100 | 0.48 | 0.005 |
| 2 | 20 | 1650.0 | 550 | 1500 | 0.30 | 0.005 |
| 3 | 20 | 2000.0 | 1200 | 3120 | 0.25 | 0.005 |
| Bedrock | - | 2438.3 | 1600 | 3820 | - | - |

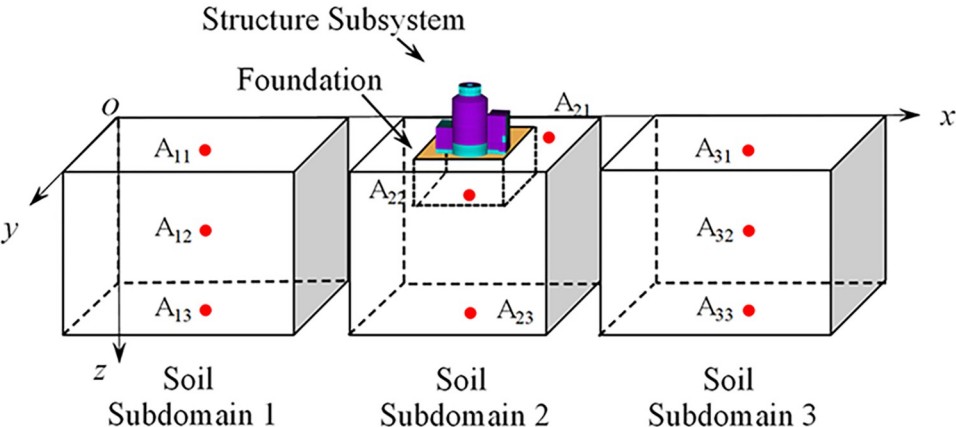

**Fig 16. Schematic diagram of the partitioned analysis of soil-structure interaction and the position of the reference points of the soil subdomain.**

**Analysis of calculation results.** The input seismic motion was specified at the bedrock and propagates vertically as SV waves (with the *x*-direction as the primary response direction), which were utilized to calculate the free-field response.

It is important to highlight that when the incident SV wave angle was below a certain threshold, both the SV wave and the seismic wave will be reflected at the soil interface simultaneously. Furthermore, there will be reflections of both SV and P waves at the soil interface when the incident SV wave angle was less than a specific value. In scenarios where the incident SV wave angle was smaller than this critical angle, reflections of both SV and P waves occur at the soil interface. Conversely, when the incident SV wave angle exceeds the critical angle, only the reflected SV wave was present at the soil interface. The critical angle, denoted as $\theta_{cr}$, was directly related to Poisson's ratio and can be determined using the following equation:

$$\theta_{cr} = \arcsin(V_s/V_p) \qquad (25)$$

In this example, the soil layer selection comprises a total of three layers (as outlined in Table 2): the first layer of soil critical angle was calculated as $\theta1_{cr} = \arcsin(330/2100) \approx 9.05°$, the second layer of soil critical angle was calculated as $\theta2_{cr} = \arcsin(550/1500) \approx 21.51°$, and the critical angle of the third layer of soil was calculated as $\theta3_{cr} = \arcsin(1200/3120) \approx 22.62°$. Through comprehensive analysis of the oblique angle of incidence of the space, the selected values have been validated as feasible through comparative study. In this study, the displacement response of the nuclear power structure and the soil (foundation) was examined under the incidence of SV waves at space angles of ($\theta_1 = 0°, \theta_2 = 20°$), ($\theta_1 = 20°, \theta_2 = 0°$), and ($\theta_1 = 20°, \theta_2 = 20°$), respectively. The displacement time histories of the relevant reference points were presented below.

(1) **Response of the soil**

Reference points $A_{11}$ and $A_{31}$ represent the nodes located on the surface of soil subdomain 1 and soil subdomain 3, respectively (as depicted in Fig 16). In Fig 17, it can be observed that under the incidence angle of ($\theta_1 = 20°, \theta_2 = 0°$), a comparison of the results indicates a time difference of approximately 0.06s at the onset of wave vibration. In this study, with a model soil layer thickness of 60m, where $\theta_1 = 20°$, and an average vertical propagation velocity of approximately 1000m/s for the three soil layers, calculations demonstrate that it takes roughly

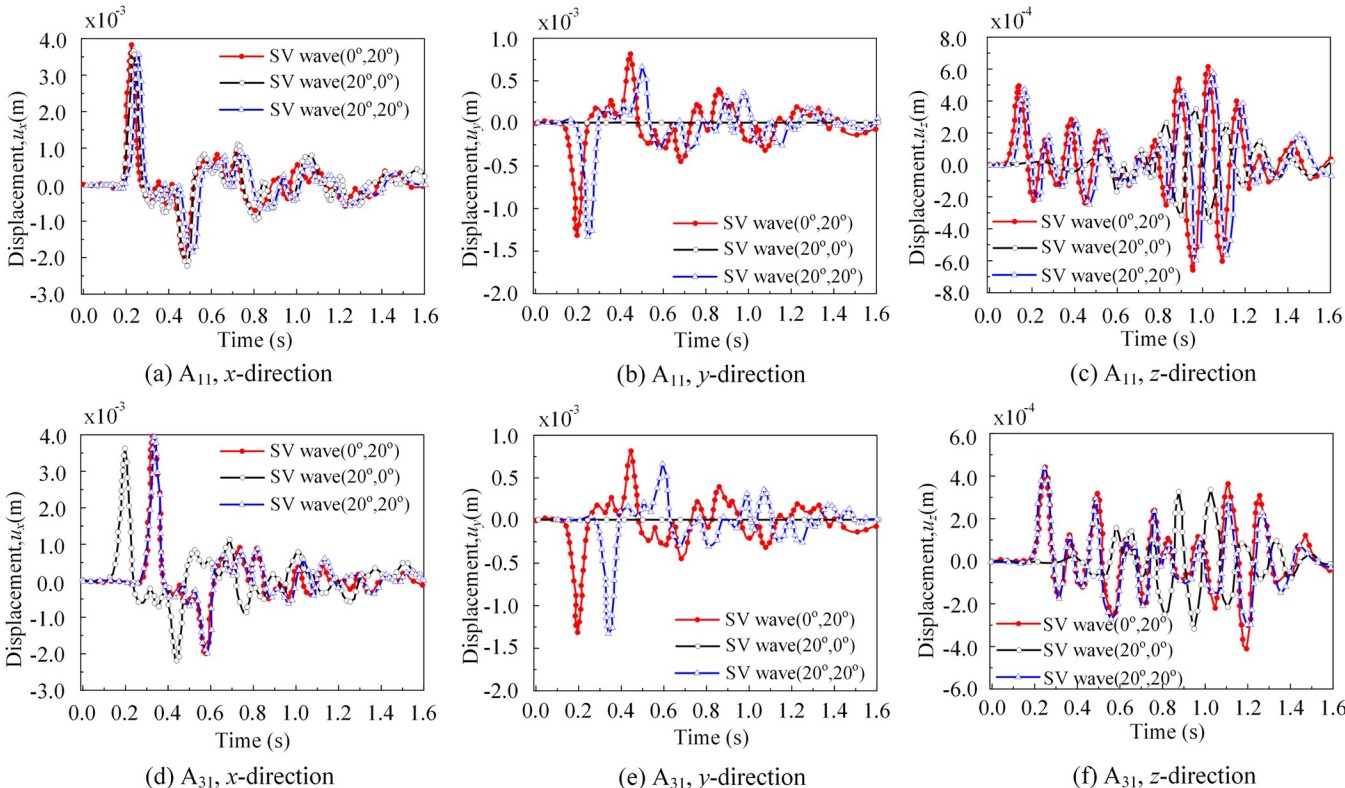

**Fig 17. Displacement response of soil reference points $A_{11}$ and $A_{31}$.** (a) Displacement response of soil reference point $A_{11}$ in the $x$-direction. (b) Displacement response of soil reference point $A_{11}$ in the $y$-direction. (c) Displacement response of soil reference point $A_{11}$ in the $z$-direction. (d) Displacement response of soil reference point $A_{31}$ in the $x$-direction. (e) Displacement response of soil reference point $A_{31}$ in the $y$-direction. (f) Displacement response of soil reference point $A_{31}$ in the $z$-direction.

0.06s for the wave to vertically propagate to the surface of the soil body. Remarkably, the simulation results obtained through finite element analysis align well with the theoretical solution.

In Fig 18, reference point $A_{13}$ is located on the bottom surface of soil subdomain 1, while reference point $A_{33}$ is located on the bottom surface of soil subdomain 3. These two reference points are situated at the same positional height but in different soil subdomains. Upon analyzing the displacement response under an incident angle of $\theta_2 = 0°$, no time difference was observed. However, when the second spatial incidence angle $\theta_2 \neq 0°$, specifically for the cases with spatial incidence angles of ($\theta_1 = 0°$, $\theta_2 = 20°$) and ($\theta_1 = 20°$, $\theta_2 = 20°$), the onset of vibration in all three directions and the time of arrival for peak displacement response were delayed by approximately 0.08s. This phenomenon can be attributed to the wave vibration plane initiating from the left front corner point of the site model. When there was an inclination angle of 20°, the wave propagation time from the reference point at the bottom of soil subdomain 1 to the reference point at the bottom of soil subdomain 3 was approximately 0.08s, as predicted by theoretical calculations. This finding provides preliminary verification of the accuracy of our method.

In Fig 19, upon comparing all reference points at the site, it was evident that the surface reference point $A_{21}$ of soil subdomain 2, which was near the foundation, experienced the maximum displacement response in the $x$-direction, reaching a value of $3.5 \times 10^{-3}$ m. This phenomenon can be attributed to two factors. Firstly, the free surface of the site amplifies the seismic wave, resulting in an increased maximum displacement typically occurring at the free

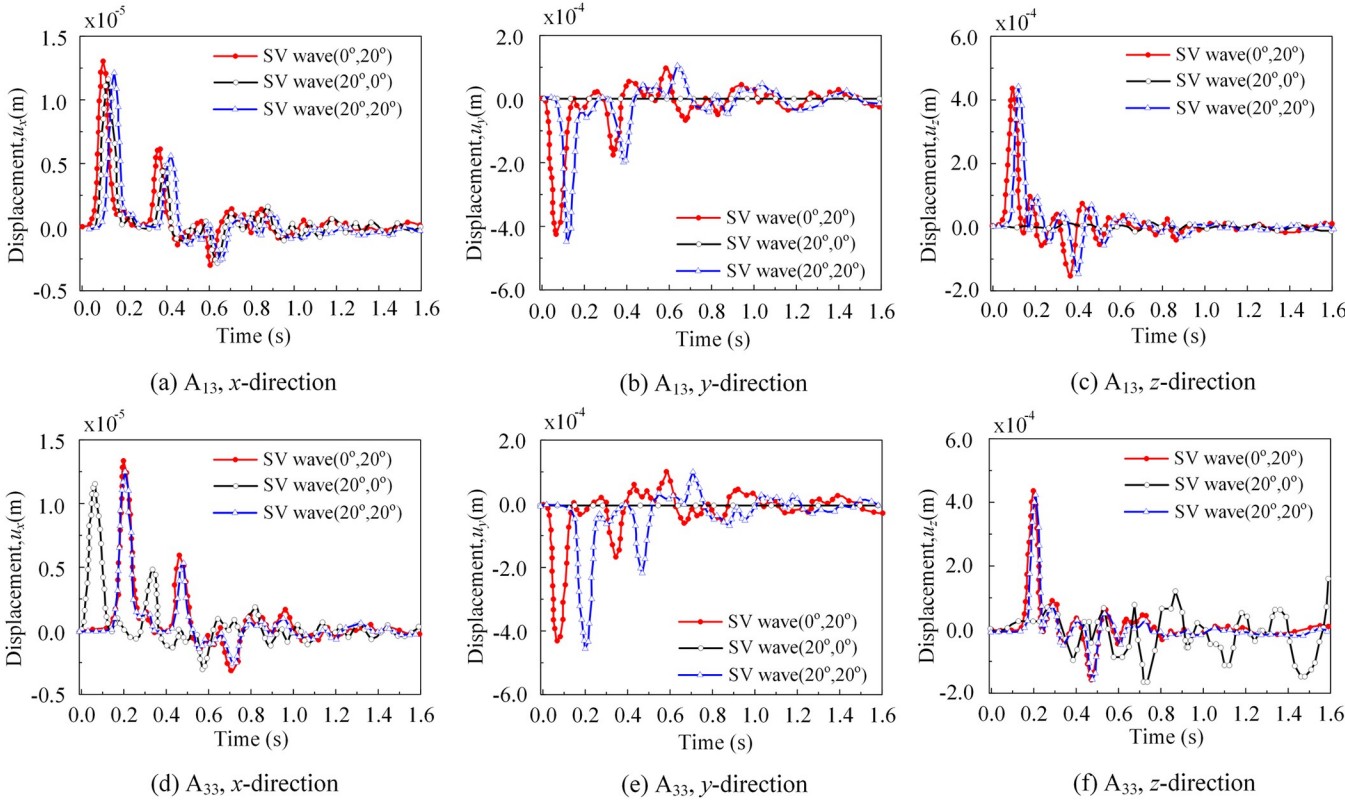

**Fig 18. Displacement response of soil reference points A₁₃ and A₃₃.** (a) Displacement response of soil reference point $A_{13}$ in the $x$-direction. (b) Displacement response of soil reference point $A_{13}$ in the $y$-direction. (c) Displacement response of soil reference point $A_{13}$ in the $z$-direction. (d) Displacement response of soil reference point $A_{33}$ in the $x$-direction. (e) Displacement response of soil reference point $A_{33}$ in the $y$-direction. (f) Displacement response of soil reference point $A_{33}$ in the $z$-direction.

surface. Secondly, in the horizontal direction, the displacement response at the point around the base (point $A_{21}$) differs significantly from the corresponding values at points away from the base (points $A_{11}$ and $A_{31}$). Point $A_{21}$ is near the superstructure and experiences the largest "quadratic nonlinear" effect due to the SSI.

For the same reference point in the $x$-direction, under three different incidence angle cases, the initial displacements occur in the following order: $T_1(20°,0°) > T_2(0°,20°) > T_3(20°,20°)$, from early too late. Considering the influence of the spatial angle, the effective wave velocity perpendicular to the $z$-direction changes, where $V_1(20°,0°) > V_2(0°,20°) > V_3(20°,20°)$, resulting in a time difference in the displacement of the same reference point. Based on the displacement response results and the theoretical method mentioned above, the calculated time for the first peak displacement was as follows: $T_1(20°,0°) < T_2(0°,20°) < T_3(20°,20°)$ ($T_1 = 0.16s$, $T_2 = 0.18s$, $T_3 = 0.2s$). Furthermore, the amplitude of displacement shows a contrast with $UF_1(20°,0°) > UF_2(0°,20°) > UF_3(20°,20°)$. It can be observed that the presence of the first spatial incidence angle $\theta_1$ significantly reduces the maximum magnitude in the $x$-direction at the reference point.

For the same reference point, when the incident angle is $(0°,20°)$, the $y$-direction displacement is negligible, almost zero. This is because when the first spatial incident angle $\theta_1$ was zero, little to no $y$-direction displacement occurs. Additionally, due to a time difference of approximately 0.07s at the surface point, the first displacement peak occurs at $T_1(20°,0°) < T_3(20°,20°)$ ($T_1 = 0.16s$, $T_3 = 0.2s$). Regarding the $z$-direction, the basic displacement pattern

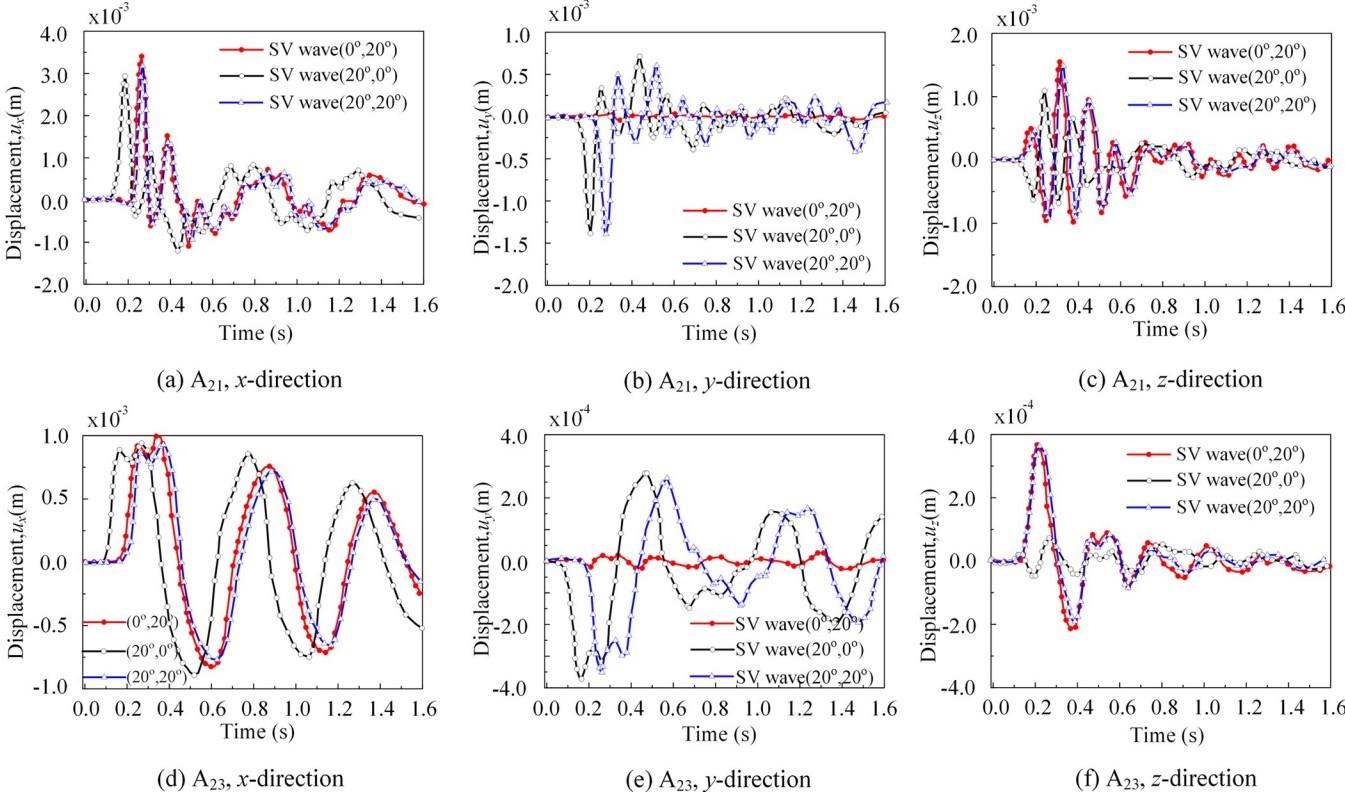

**Fig 19. Displacement response of soil reference points $A_{21}$ and $A_{23}$.** (a) Displacement response of soil reference point $A_{21}$ in the $x$-direction. (b) Displacement response of soil reference point $A_{21}$ in the $y$-direction. (c) Displacement response of soil reference point $A_{21}$ in the $z$-direction. (d) Displacement response of soil reference point $A_{23}$ in the $x$-direction. (e) Displacement response of soil reference point $A_{23}$ in the $y$-direction. (f) Displacement response of soil reference point $A_{23}$ in the $z$-direction.

was consistent with the conclusion. It should be noted that when the incident angle is $(20°,0°)$, the $z$-direction displacement is minimal at the beginning of the second half before gradually increasing. The first displacement peak occurs at $T_2(0°, 20°)$ (0.115s), which was slightly earlier than $T_3(20°, 20°)$ (0.12s).

(2) **The response of the foundation**

From Fig 20, for the reference points on the foundation, comparing the translational and rotational displacements in the three directions of the foundation reveals that the order of the displacements from early to late was $T_1(0°,20°) > T_2(20°,0°) > T_3(20°,20°)$, respectively. Although the foundation rotation was relatively small, the height of the nuclear power super-structure was considerable, rendering the rotation of the foundation non-negligible.

(3) **The response of the structure**

As depicted in Fig 21, a comparative analysis of the time histories of translational displacements along $x$-, $y$-, and $z$-directions for reference points on the upper structure (as shown in Fig 14(B)) reveals that the displacements occur in the order of $T_1(0°,20°) > T_2(20°,0°) > T_3(20°,20°)$, from early to late, respectively. The response of the structure resembles that of a rigid foundation, since such a foundation was employed.

Given the considerable height of the structure, rotational displacements in all three directions at the structural reference point cannot be disregarded, with the displacement in the z-

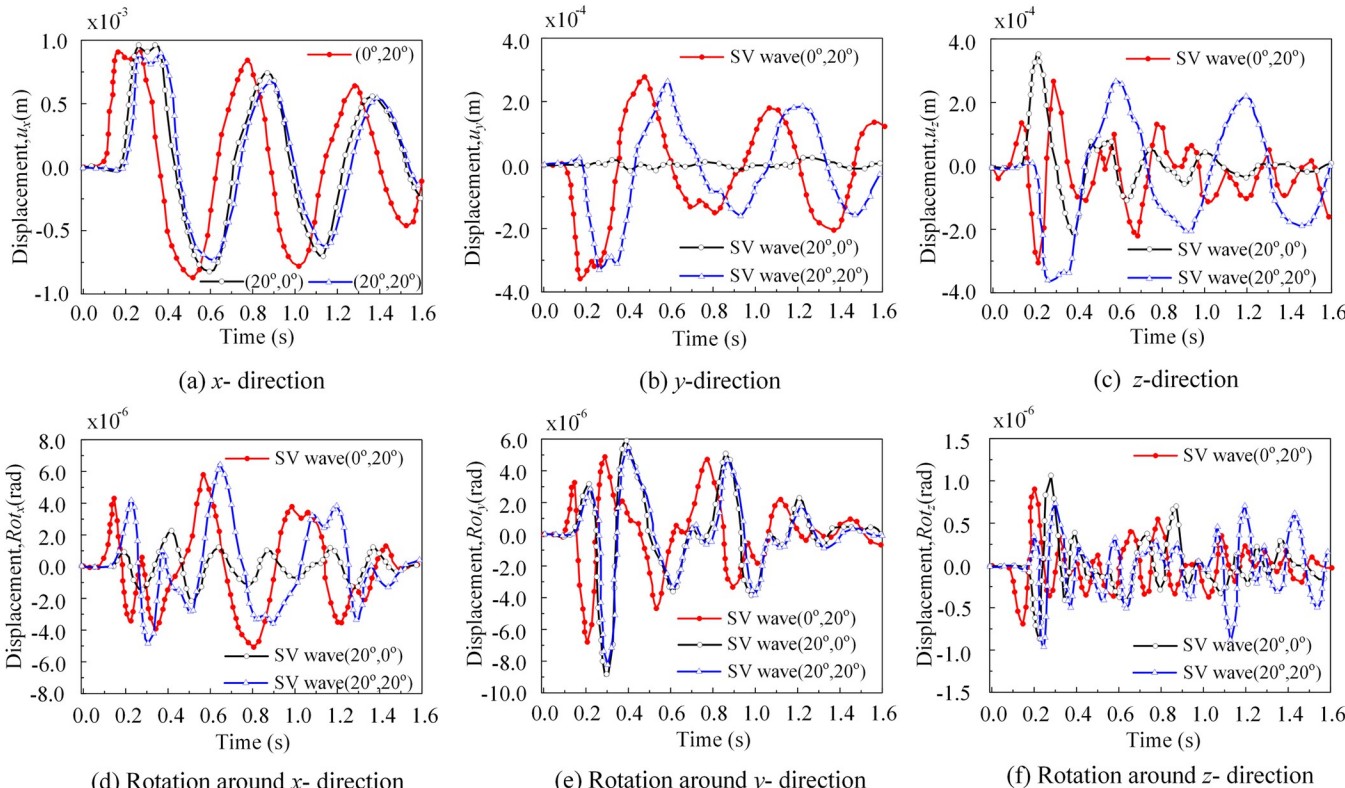

**Fig 20. Displacement response of the foundation reference point A_{22}.** (a) Displacement response of foundation reference point $A_{22}$ in the $x$-direction. (b) Displacement response of foundation reference point $A_{22}$ in the $y$-direction. (c) Displacement response of foundation reference point $A_{22}$ in the $z$-direction. (d) Displacement response of foundation reference point $A_{22}$ rotation around $x$-direction. (e) Displacement response of foundation reference point $A_{22}$ rotation around $y$-direction. (f) Displacement response of foundation reference point $A_{22}$ rotation around $z$-direction.

direction being slightly smaller compared to those in the $x$-and $y$-directions. The reference point at the top of the structure (point 211365) and the reference point at the bottom of the structure (point 47096) exhibit an inverse time difference of approximately 0.02s, which is consistent with real-life scenarios. Moreover, it can be observed that the three translational displacements of the reference point at the bottom of the structure (point 47096) were mostly in agreement with the displacement time-course curves in the three directions of the base reference point, whereas the displacement time-course curves of the reference point at the top of the structure (point 211365) demonstrate greater differences. Assuming elastic soil conditions and tied soil-foundation interfaces, the study demonstrates that the rocking vibration mode of the soil-structure system was amplified by the presence of the auxiliary building through a detrimental out-of-phase rotational interaction mechanism. This effect was heightened along the direction of the structure's height and may further be impacted by the whiplash effect. The accuracy of our proposed algorithm was validated by the arithmetic example.

## Seismic response analysis of a deep-water bridge

In this section, we aim to investigate the impact of site-structure coupling, non-uniform excitation, and SSI on the seismic response of a bridge located in a seismically active region. Specifically, we explore the influence of varying seismic wave incidence angles on the bridge's behavior. Given the bridge's proximity to a water body, it was crucial to analyze its response to

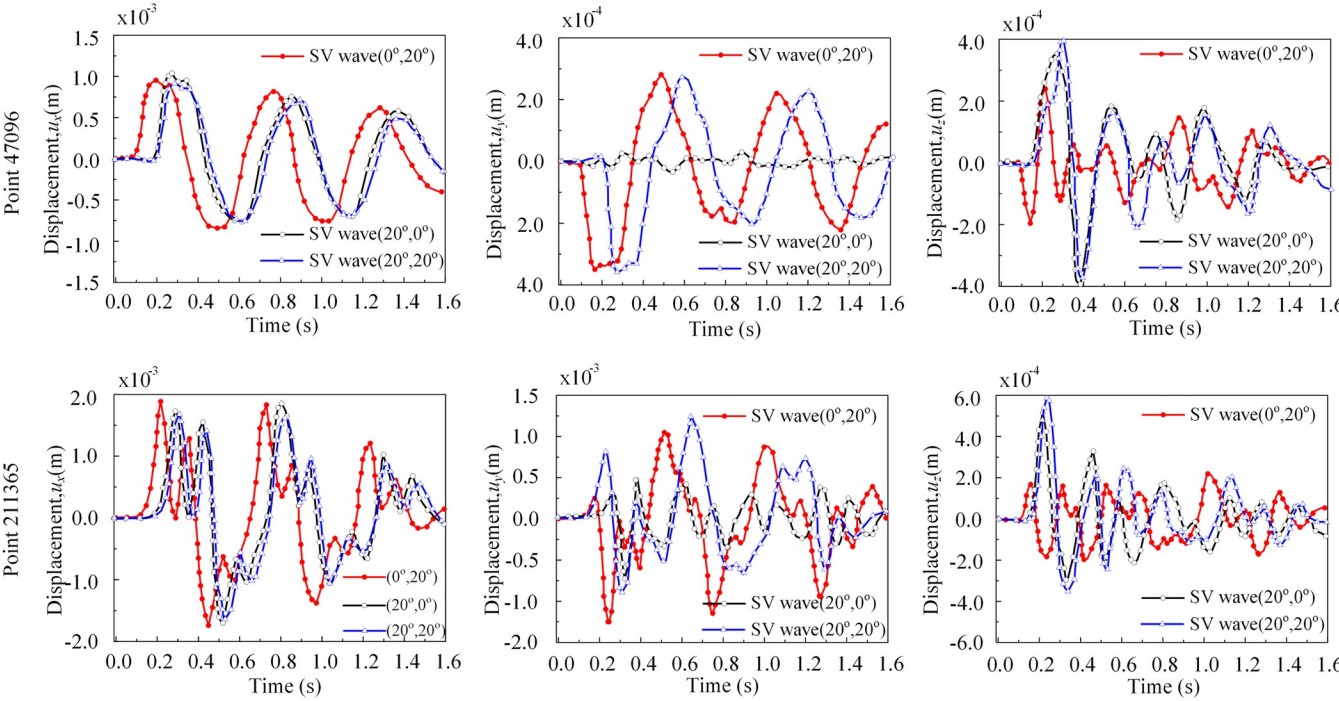

**Fig 21. The response of the structure.** (a) The response of the reference point at the top of the structure (point 211365) in the x-,y-, and z- directions. (b) The response of the reference point at the top of the structure (point 47096) in the x-,y-, and z- directions.

seismic action while considering the presence of water and assessing the extent to which the water body affects the structure's seismic response.

**Model and parameters.** This study takes a multi-span continuous girder bridge located in a specific area as an example. The site conditions, heights of pile foundations, and bridge piers exhibit variations along the longitudinal direction of the bridge. However, for simplicity, these differences were disregarded, and the soil was assumed to be layered. In terms of adjacent bridge spans, only the weight of the girder was taken into consideration. The dimensions of

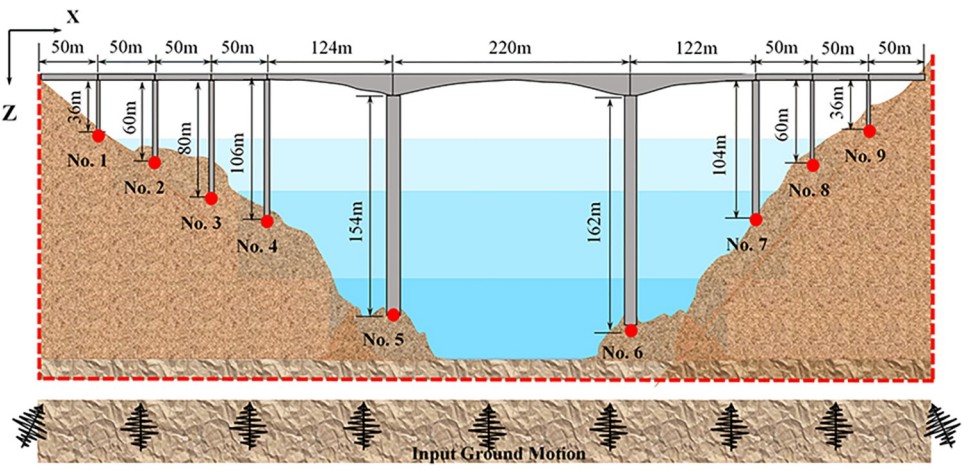

**Fig 22. General layout of the bridge.**

**Table 3. Material properties.**

| Part | Mass density ($10^3$kg/m$^3$) | Poisson ratio | Elastic modulus (pa) | Shear wave velocity $V_s$(m/s) | Compressional wave velocity $V_p$ (m/s) |
|---|---|---|---|---|---|
| Main girder | 2.50 | 0.20 | $3.60\times10^{10}$ | 2450 | 4000 |
| Piers, T-beams | | | $3.45\times10^{10}$ | 2398 | 3916 |
| Pile-cap | | | $3.02\times10^{10}$ | 2244 | 3664 |

the bridge and its main sections were illustrated in Fig 22. The bridge consists of ten spans, with lengths of 50 m, 50 m, 122 m, 124 m, and 220 m, resulting in a total length of 816 m. The main bridge was a continuous rigid structure, with a maximum span of 220 m. On the other hand, the approach bridge was a simply supported girder with a span of 50 m. The main bridge was characterized by a variable section continuous rigid structure, featuring a single box and a single chamber main girder section made of C60 concrete.

In Fig 22, the pier heights for Pier No. 5 and Pier No. 6 of the main bridge abutment were 154 m and 162 m, respectively. These piers are constructed using reinforced concrete rectangular hollow sections, with C50 concrete specifications. The bearing platform beneath the main bridge abutment has a height of 6 m, a plan dimension of 18.0 m × 14.0 m, and is supported by a 4 × 3 arrangement of square piles measuring 2.0 m × 2.0 m. Both the bearing platform and the pile foundation utilize C30 concrete specifications. The junction pier also employs a reinforced concrete rectangular hollow pier, with Pier No. 4 being 106 m in height and Pier No. 7 being 104 m in height. The bridge pier has a height of 4.0 m and a plan dimension of 10.0 m × 14.0 m. The foundation consists of a 2 × 3 arrangement of 2.0 m × 2.0 m square piles, with the concrete specifications for both the bearing platform and the pile foundation being C30. As for the approach piers, they were constructed using reinforced concrete rectangular solid sections, with a concrete specification of C50. The bridge pier has a height of 2.0 m and a plan dimension of 10.0 m × 10.0 m. The foundation comprises a 2 × 2 arrangement of 2.0 m × 2.0 m square piles, with the abutment and the pile foundation utilizing C30 concrete specifications. Table 3 displays the material parameters of the bridge.

The three-dimensional finite element model of the bridge was implemented using the ANSYS commercial finite element software, as depicted in Fig 23(A). The main girders of both the main bridge and approach bridge were simulated using BEAM188 girder elements, while the piers, cover girders, and abutments were simulated using SOLID185 hexahedron eight-node solid elements.

The BEAM188 element incorporates the Timoshenko beam structure theory, which accounts for shear deformation effects. This element was suitable for analyzing main girders with variable sections and T-section girders. The variable section segments were modeled using the SECTYPE command in the APDL language. Coupling constraints between the piers and girders were applied at the bridge supports using CE and CERIG commands in ANSYS APDL language. As shown in Fig 23(B) and 23(C), the simply supported girders were coupled in the corresponding directions via CE commands, and at the continuous stiffeners, rigid regions were generated using the CERIG command.

**Input seismic wave.** Our primary focus is to investigate the impact of site-reservoir-bridge interaction on the seismic response of the structure. Given that the influence of moving water on the structure under vertical excitation is limited, we have chosen to design the acceleration response spectrum in the horizontal direction.

Referring to the "Specifications for Seismic Design of Highway Bridges" (JTG/T 2231-01-2020, China) and the "Seismic ground motion parameters zonation map of China" (GB 18306–2015, China), we have determined that the characteristic period of the site's class II category is $T_g$ = 0.45s, and the horizontal peak basic ground vibration acceleration is A = 0.15 g.

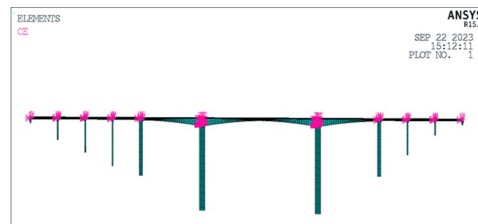

(a) Integral structural models.

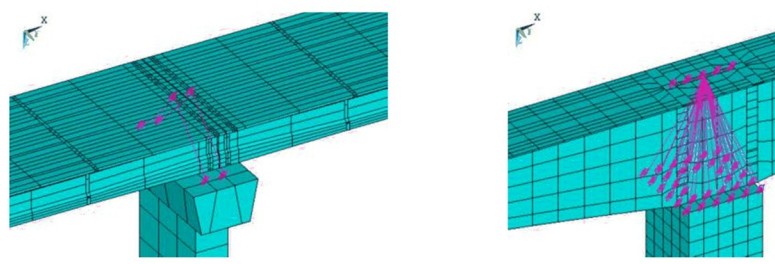

(b) Simply supported beam pier coupling.   (c) Continuous rigid beam-pier coupling.

**Fig 23. Finite element model of the bridge.** (a) Integral structural models. (b) Simply supported beam pier coupling. (c) Continuous rigid beam-pier coupling.

Considering the geological conditions of the site, which was classified as $I_0$, adjustments have been made to the characteristic period $T_g$ based on the Code. In this paper, we conduct the first stage of seismic analysis, specifically the E1 seismic action, to evaluate the seismic response of the bridge. The correlation coefficients for the design acceleration response spectrum were presented in Table 4. Following a screening process, seismic waves were selected and amplitude-modulated to satisfy the specified conditions. The response spectra of these waves and the corresponding design acceleration response spectra were illustrated in Fig 24. Based on the modal analysis of the bridge, the first few modes of the structure primarily occur in the descending section of the response spectrum. As per specifications, the absolute error between the seismic wave acceleration response spectra in the structural cycle values corresponding to the response spectrum amplitude and the design acceleration response spectra amplitude should not exceed 0.01 g. Table 5 illustrates the errors in the seismic wave acceleration response spectra for the corresponding cycles.

The seismic wave utilized in this study is sourced from the San Onofre Nuclear Power Station in Southern California, USA, and records the earthquake that transpired on April 9, 1968, at Borrego Mountain. It was referred to as Borrego-San Onofre-19680409 and follows a nomenclature of place of seismic, station of record, and date of seismic. To meet the specifications of the design acceleration response spectrum, the seismic wave's acceleration amplitude was processed, and its peak acceleration was adjusted to 0.09 *g*. The seismic wave had a duration of 45.2 *s*, and its ground motion and fourier amplitude spectrum were presented in Fig 25. In this paper, the example seismic wave input originates from the left front corner

**Table 4. The design acceleration response spectrum correlation coefficients.**

| E1 | $C_i$ | $C_s$ | $C_d$ | $A(g)$ | $S_{max}(g)$ | $T_g(s)$ |
|---|---|---|---|---|---|---|
| Horizontal | 1.0 | 0.75 | 1.0 | 0.15 | 0.281 | 0.30 |
| Vertically | 1.0 | 0.60 | 1.0 | 0.15 | 0.225 | 0.25 |

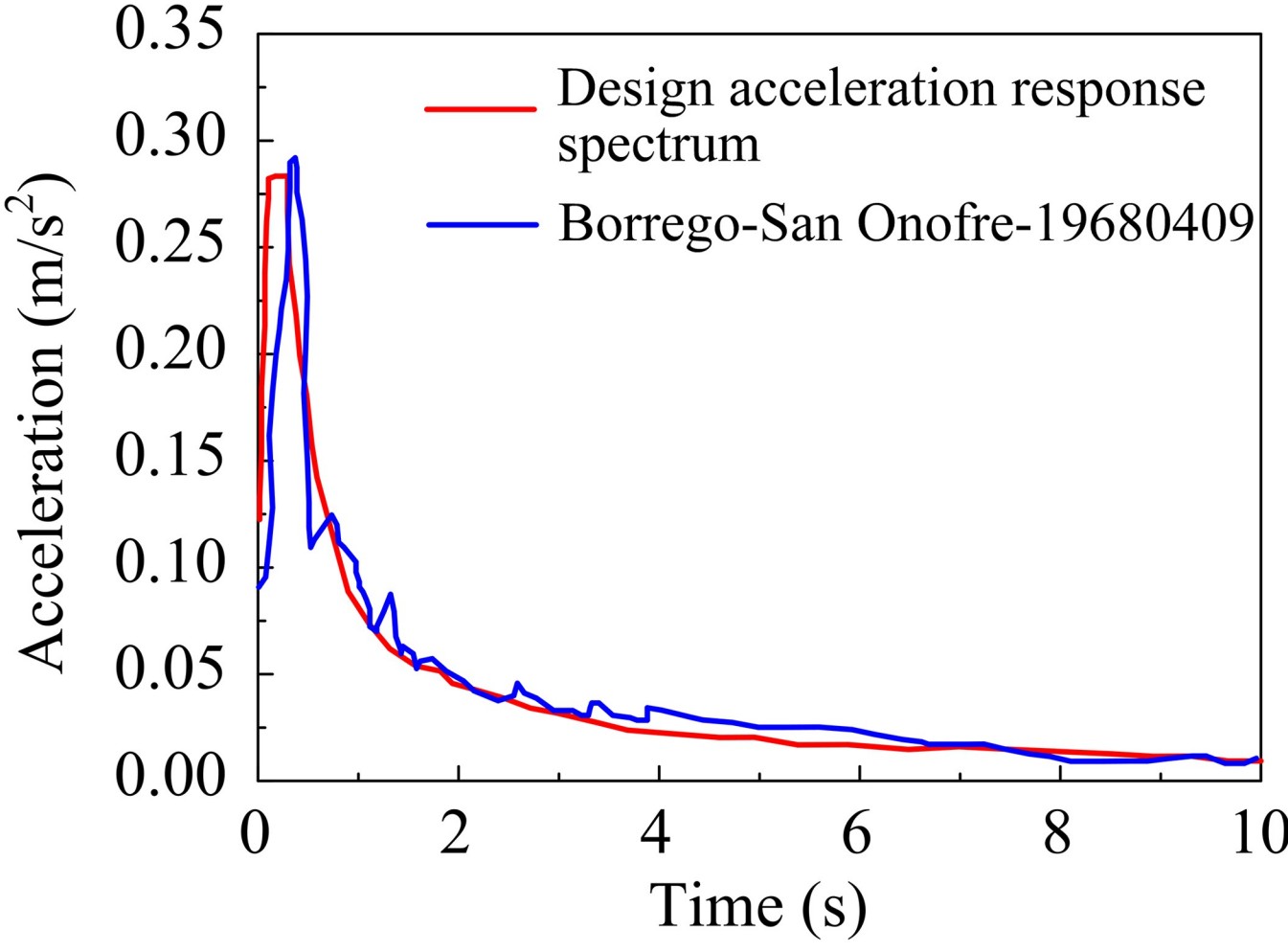

**Fig 24. Acceleration response spectrum.**

point of the finite element site, which corresponds to the location of the source, as depicted in Fig 26.

**Analysis of calculation results.** In this study, the seismic response of large-span deep-water bridges was investigated to examine the influence of relevant factors. The selected site area measures 1000m × 50m × 400m, as depicted in Fig 26. The terrain features a V-shaped canyon, and the site boundary conditions were projected boundary conditions. Table 6 presents the site-related parameters.

During earthquake response analysis of bridges situated in marine or reservoir areas, it is essential to consider site-reservoir-bridge interactions. Additionally, the traveling wave effect should not be disregarded when analyzing long-span bridge structures. Due to the large span of the bridge structure due to the ground vibration input of each support point distance is large, the seismic wave arrives at each support point, thus generating a time lag. Reflection and refraction of seismic wave propagation in the medium will generate dispersion loss, and the local soil layer of each support point will be different. Therefore, the spatial effect of seismic input should be considered in the analysis of the seismic response of large-span bridge structures [45–47].

**Table 5. Table of relative design acceleration response spectrum amplitude errors.**

| Seismic wave | T = 1.360s | T = 2.495s | T = 3.494s | T = 4.508s | T = 5.591s | T = 5.897s | T = 10.00s |
|---|---|---|---|---|---|---|---|
| W1 | +0.0001g | +0.003g | +0.005g | +0.002g | -0.0006g | -0.0002g | -0.006g |

To investigate the effect of changes in the incident angle of seismic waves on the seismic response of a large-span deep-water bridge, as shown in Fig 26. Here, three working conditions were designed, and the left, right, front, rear, and bottom surfaces of the site for each condition were transmissive artificial boundaries. Where the site material was all bedrock (as shown in Table 6), the water depth was 200 m (full reservoir), and the incident angles of seismic waves $\theta_1$ were 0° (CASE1), 10° (CASE2), and 20°(CASE3), respectively, and $\theta_1$ was 0°.

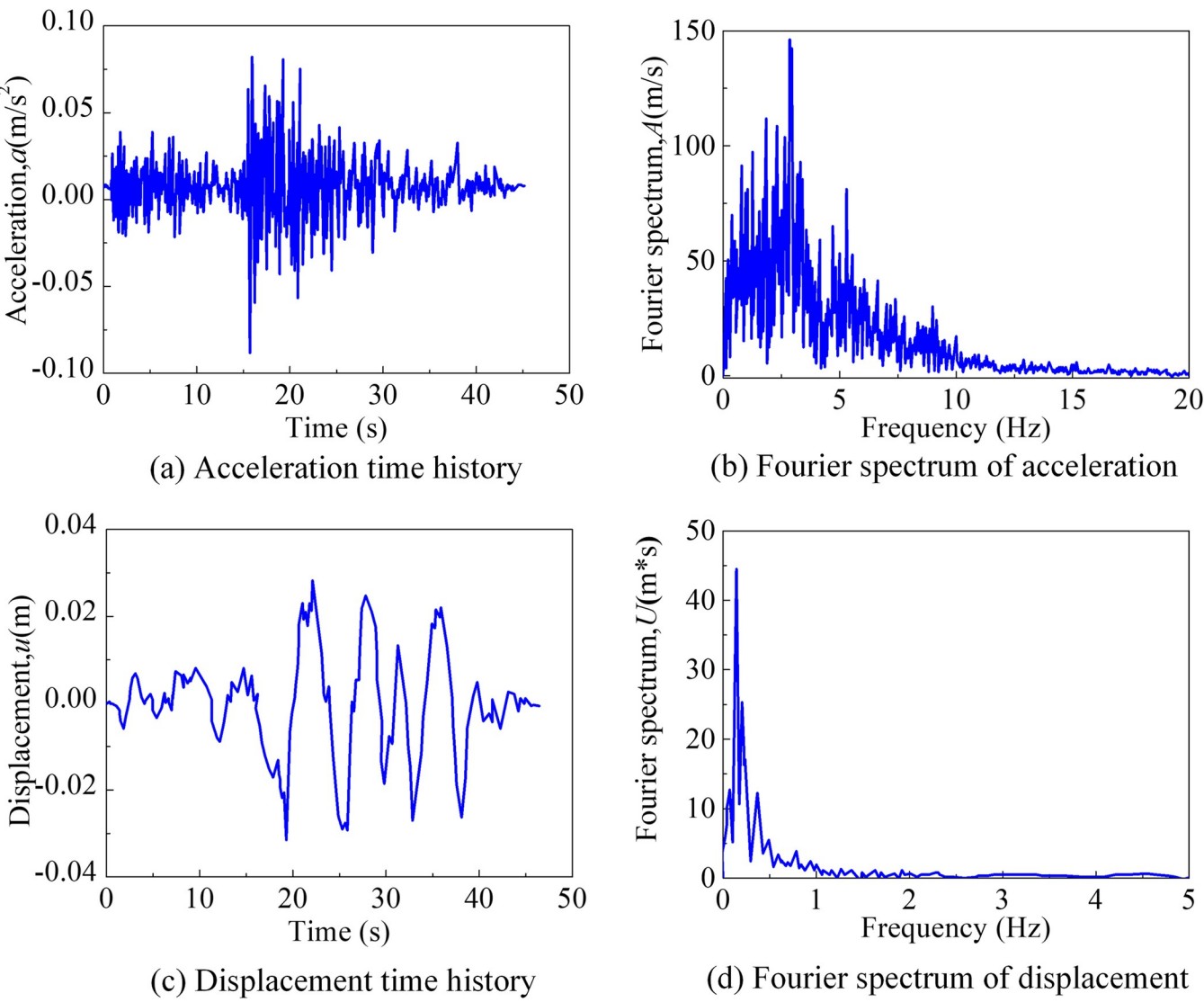

(a) Acceleration time history

(b) Fourier spectrum of acceleration

(c) Displacement time history

(d) Fourier spectrum of displacement

**Fig 25. Ground motion and its Fourier amplitude spectrum.** (a) Acceleration time history of the ground motion. (b) Fourier spectrum of acceleration of the ground motion. (c) Displacement time history of the ground motion. (d) Fourier spectrum of displacement.

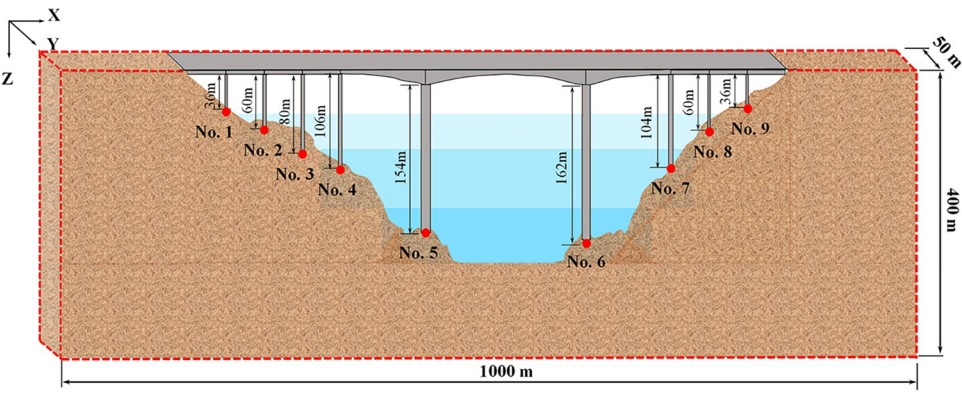

**Fig 26. Schematic diagram of the calculation site.**

**(1) The response of the structure**

As shown in Fig 27, the displacement timescale and displacement spectrum of the reference point at the bottom of the bridge Abutment No. 9 along the *x*-direction. From the figure, with the change of seismic wave incidence angle, the displacement of the reference point at the bottom of Pier No. 9 has an obvious delay and decrease trend.

**Table 6. Parameters of the site model.**

| Material | Elastic modulus (pa) | Poisson ratio | Mass density (kg/m³) | Shear wave velocity $V_s$(m/s) | Compressional wave velocity $V_p$ (m/s) |
|---|---|---|---|---|---|
| Reservoir water | / | / | 1000 | / | 1500 |
| Site | $4.50 \times 10^{10}$ | 0.2 | 2400 | 1550 | 2901 |

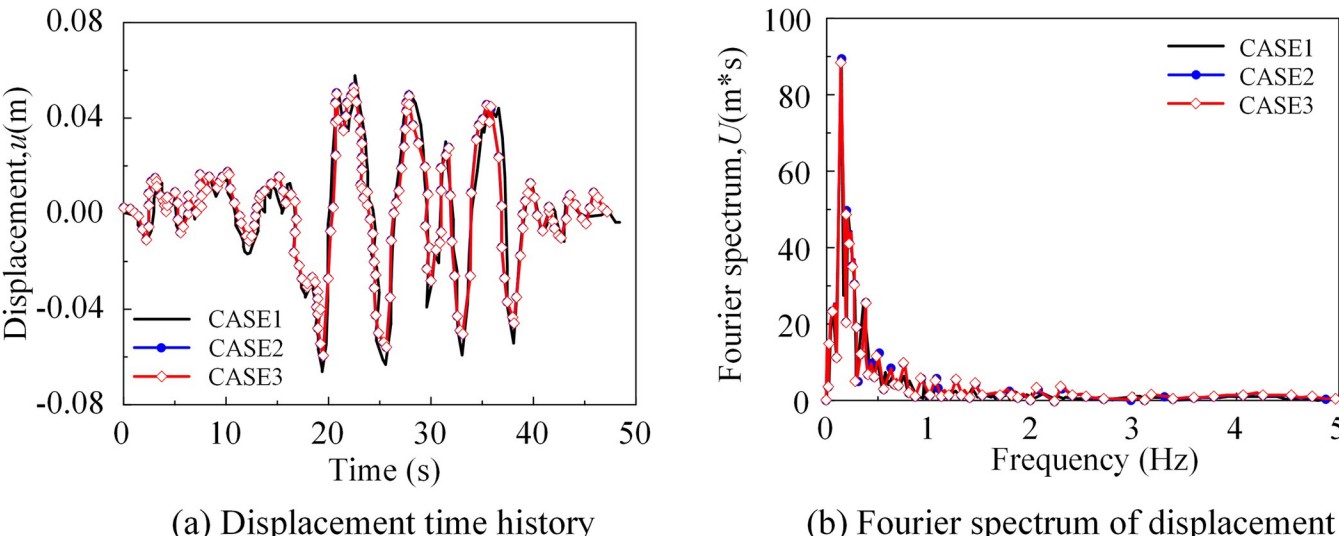

(a) Displacement time history (b) Fourier spectrum of displacement

**Fig 27. The response at the bottom of pier 9 (*x*-direction).** (a) Displacement at the bottom of pier 9 (*x*-direction).(b) Fourier spectrum of displacement at the bottom of pier 9 (*x*-direction).

**Table 7. Peak displacement response at the bottom of bridge piers under the action of seismic waves.**

| Abutment No. | Peak displacement in the downstream direction (m) | | | Peak Displacement Arrival Time (s) | | |
|---|---|---|---|---|---|---|
| | CASE1 | CASE2 | CASE3 | CASE1 | CASE2 | CASE3 |
| 1 | 0.0652 | 0.0671 | 0.0681 | 12.765 | 12.760 | 12.750 |
| 2 | 0.0649 | 0.0667 | 0.0677 | 12.765 | 12.760 | 12.750 |
| 3 | 0.0645 | 0.0662 | 0.0672 | 12.765 | 12.760 | 12.750 |
| 4 | 0.0643 | 0.0660 | 0.0669 | 12.760 | 12.760 | 12.750 |
| 5 | 0.0634 | 0.0644 | 0.0655 | 12.755 | 12.765 | 12.760 |
| 6 | 0.0635 | 0.0625 | 0.0615 | 12.755 | 12.805 | 12.835 |
| 7 | 0.0649 | 0.0631 | 0.0615 | 12.765 | 12.825 | 12.875 |
| 8 | 0.0654 | 0.0634 | 0.0616 | 12.765 | 12.835 | 12.890 |
| 9 | 0.0657 | 0.0635 | 0.0617 | 12.770 | 12.845 | 12.900 |

The peak displacement and arrival time of the reference point at the bottom of each pier along the $x$-direction under the seismic wave was depressed in Table 7. The arrival times of the peak displacements were basically in line with the previously predicted order. From the data in the table, in the $x$-direction, the peak displacement at the bottom of the near-side abutment increased by 4.5%, while the peak displacement at the bottom of the far-side abutment decreased by 6.0%. The peak displacement in the $x$-direction at the bottom of the bridge abutment under seismic wave action was shown in Fig 28. The peak displacement along the $x$-direction increases with the increase of the incident angle, the displacement of the pier bottom on its near-source side increases, and the displacement of the pier bottom on the far-source side decreases, and the non-uniform excitation was more obvious.

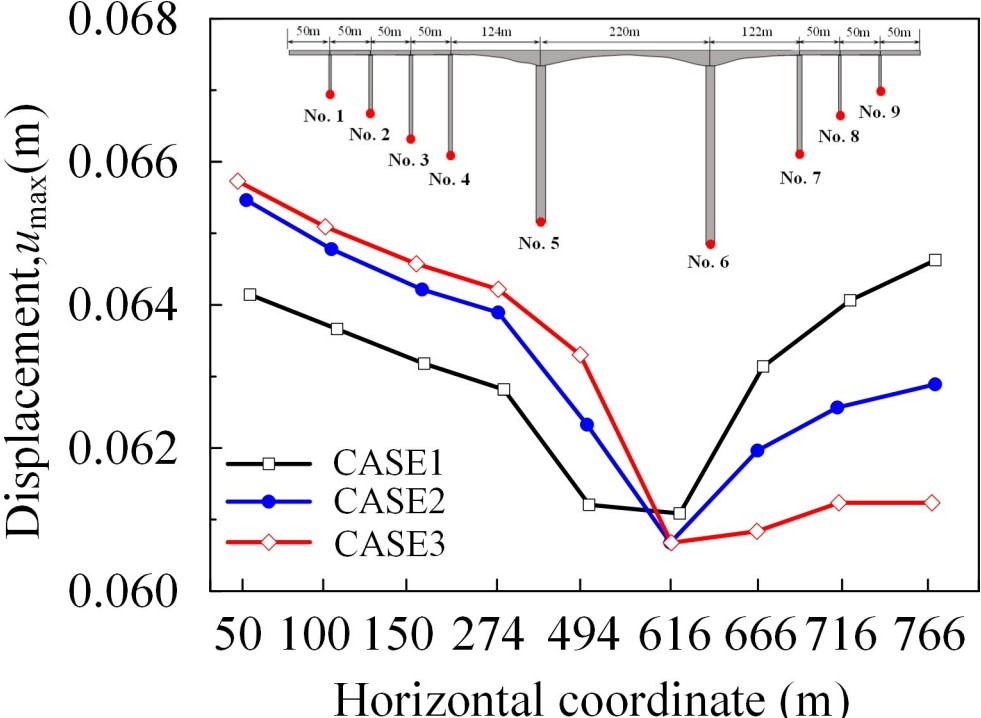

**Fig 28. Peak displacement response at the bottom of the abutment (x-direction).**

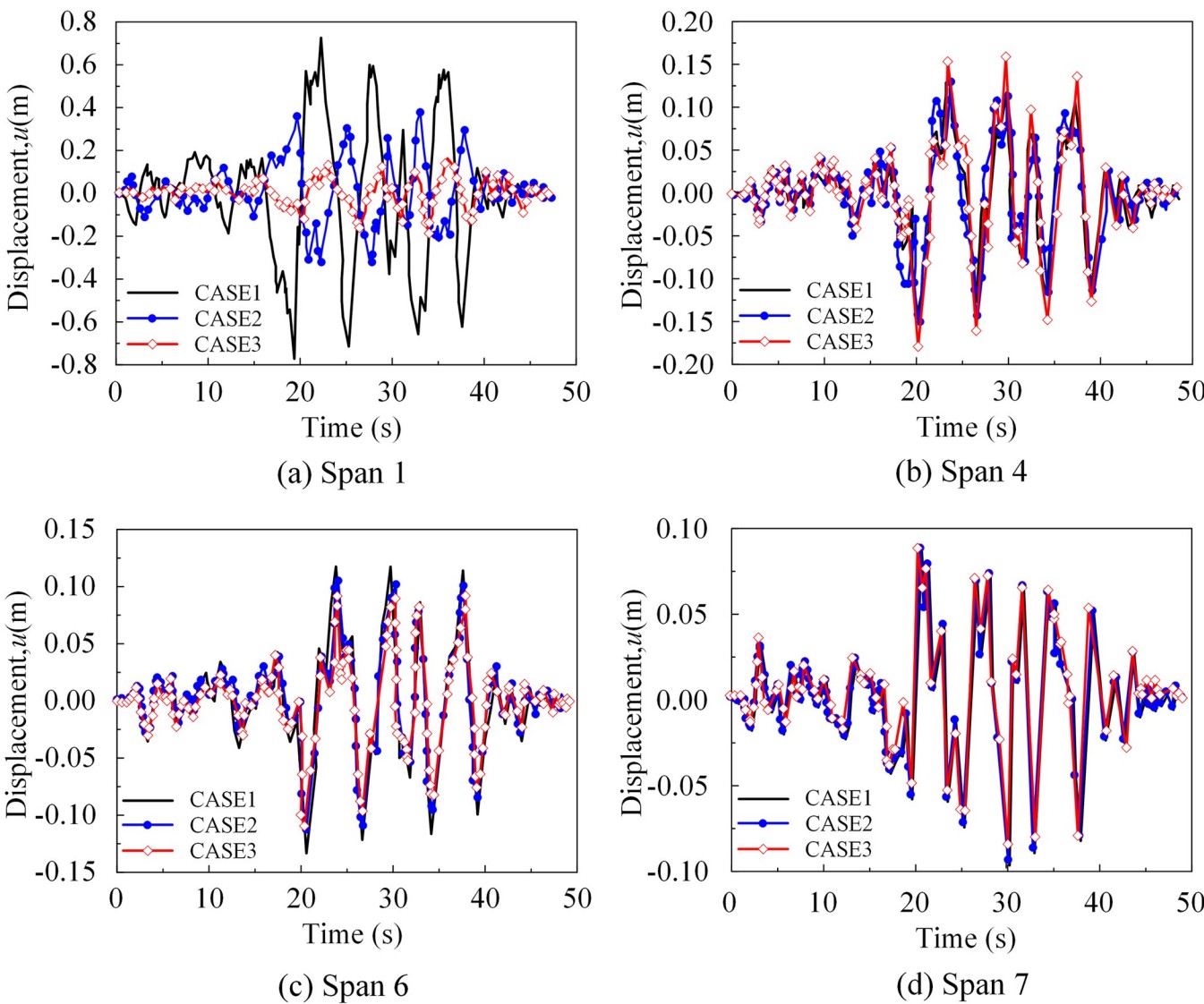

**Fig 29. Displacement response of approach bridge (*x*-direction).** (a) The time history of span 1 displacement at the *x*-direction. (b) The time history of span 4 displacement at the *x*-direction. (c) The time history of span 6 displacement at the *x*-direction. (d) The time history of span 7 displacement at the *x*-direction.

Fig 29 presents the time history of main girder displacement at the *x*-direction support of the approach bridge. As shown in the figure, the displacement response of the main girder at the far-source side support was identical to that at the bottom of its abutment, and it decreases with an increase in the incident angle (Fig 29(C) and 29(D)). However, the near-source side approach bridge exhibits a distinct trend from the response at the bottom of its abutment, owing to the influence of the dynamic characteristics of the abutment (Fig 29(A)).

Fig 30 illustrates the corresponding displacement at the main girder support of the approach bridge under seismic wave excitation. The peak displacement at the support of the approach bridge reduces as the incidence angle approaches perpendicular to the canyon slope. This outcome may be attributed to non-uniform excitation at the foundation of the bridge abutments caused by the oblique incidence and the site-reservoir-bridge interaction, leading to varying response results.

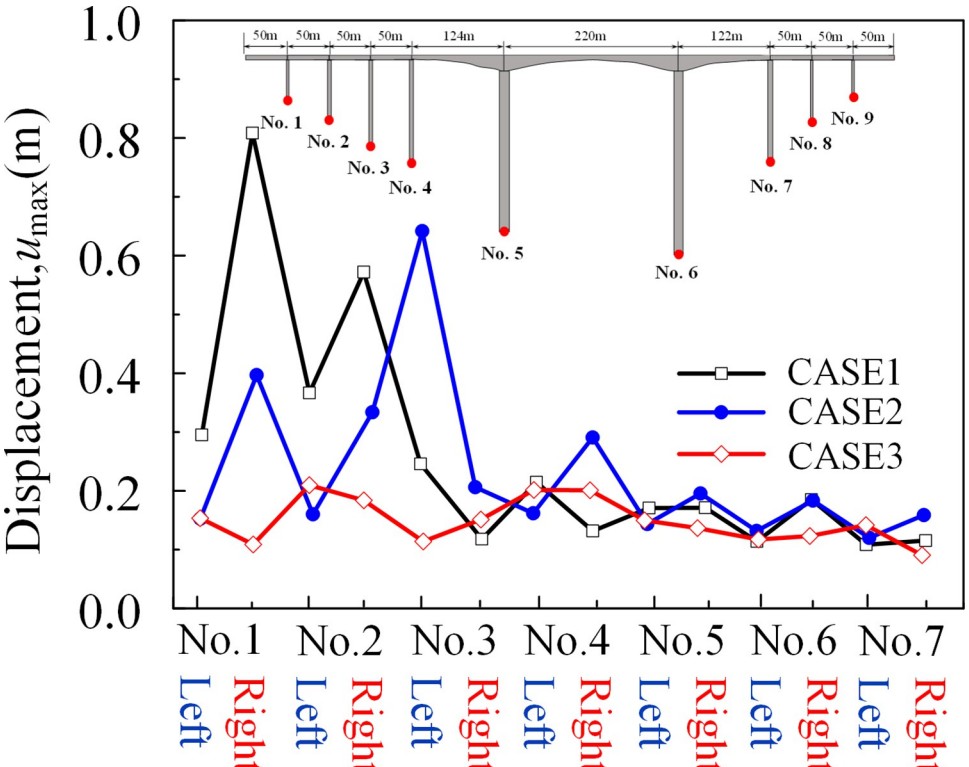

**Fig 30. Peak displacement response at the bearing of the main girder of the approach bridge (*x*-direction).**

The time history of main girder displacement at the *x*-direction support and pier-girder consolidation was depicted in Fig 31. The findings are akin to those of the approach bridge supports, but there were variations in response in individual piers due to factors such as self-oscillation frequency (Fig 31(A)).

(2) **Structural Internal Force Response**

Fig 32 illustrates the time-history curve of the shear force along the *x*-direction at the foundation of the bridge pier. From the figure, it can be observed that the shear force at the bottom of the near-side piers increases with the incident angle (Fig 32(A)), whereas the shear force at the bottom of the far-side piers decreases with the incident angle (Fig 32(B)).

The peak shear force at the bottom of each pier under seismic wave action was depicted in Fig 33. As shown in the figure, Piers 5 and 6 were significantly affected by the reservoir water due to their larger immersion depth, resulting in distinct variations under complex interaction. Notably, the peak shear force at the bottom of Pier 5 exhibited a 26.8 percent increase compared to the vertical incidence.

Based on the results, it can be observed that, in the absence of considering site nonlinearity and bridge structure nonlinearity, the response at the bottom of the abutment remains unaffected by the increase in reservoir depth. However, the shear force at the bottom of the abutment tends to decrease. The shear force at the reference point, located at the bottom of Abutment *N*o. 5, exhibits an initial increase followed by a subsequent decrease, which can be attributed to the dynamic characteristics of the abutment.

The responses and internal forces at the bottom of the abutments align with the anticipated changes: as the seismic wave incidence angle increases, the response of the near-source side

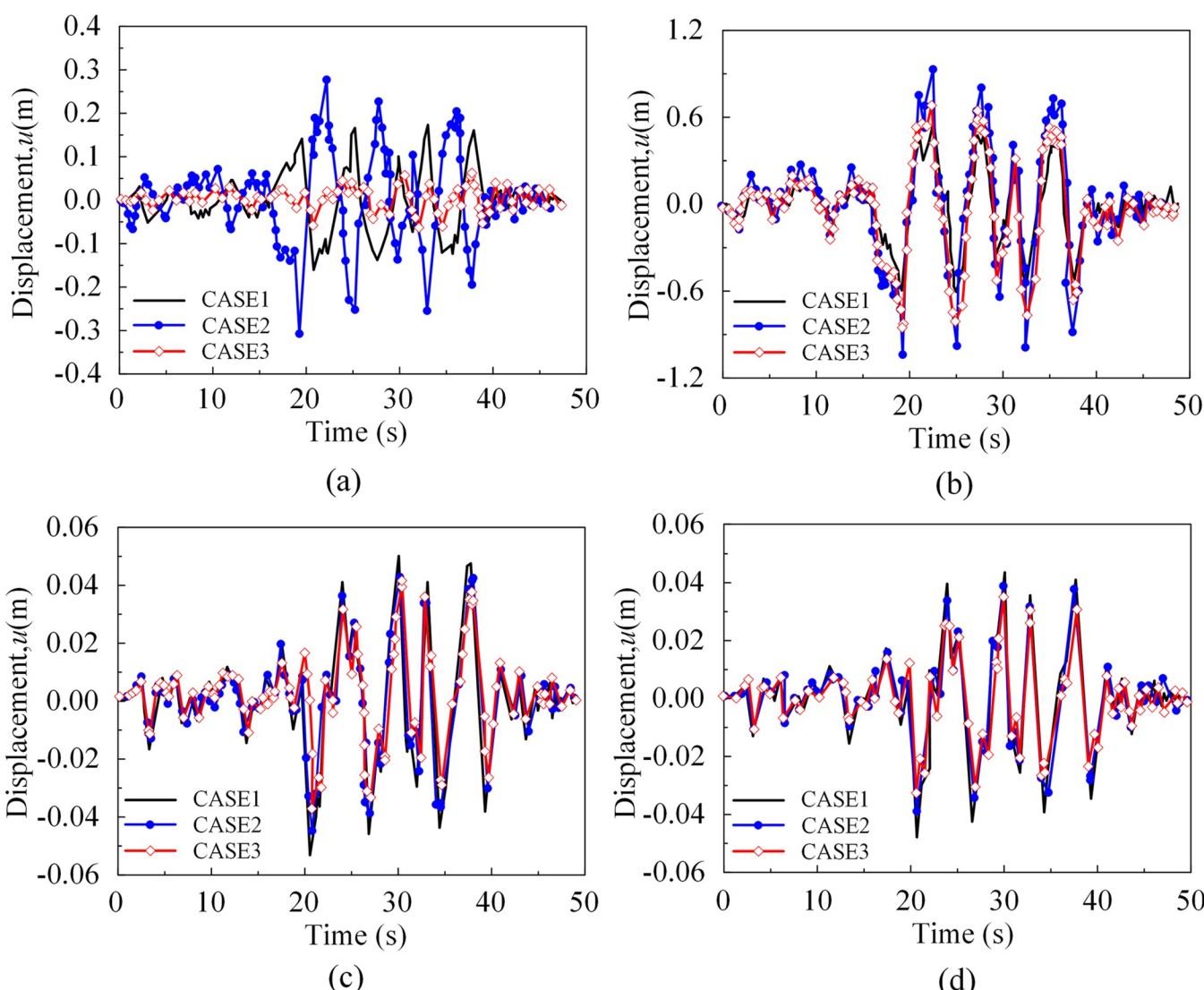

**Fig 31. Displacement response of the main girder of the main bridge (*x*-direction).** (a) Time history of displacement at the left end bearing of the main girder of the main bridge. (b) Displacement at the left end of the main girder of the main bridge at the cemented joint. (c) Time history of displacement at the right end of the main girder of the main bridge at the consolidation point. (d) Time history of displacement at the right end bearing of the main girder of the main bridge.

abutment increases while that of the far-source side abutment decreases. The displacement response of the main girder in the approach bridge decreased with the oblique incidence angle, particularly when the incidence angle was perpendicular to the valley slope, resulting in the same peak displacement response for the main girder in the approach bridge near the source. The observed responses of the bridge under seismic wave action conform to the expected patterns, thus validating the accuracy of our proposed methodology from a practical perspective.

## Conclusions

This paper presents a numerical wave analysis of the scattering of plane waves incident from any spatial direction by combining the transmission boundary and viscoelastic boundary simulations of wave scattering problems. Utilizing the transfer matrix method and coordinate

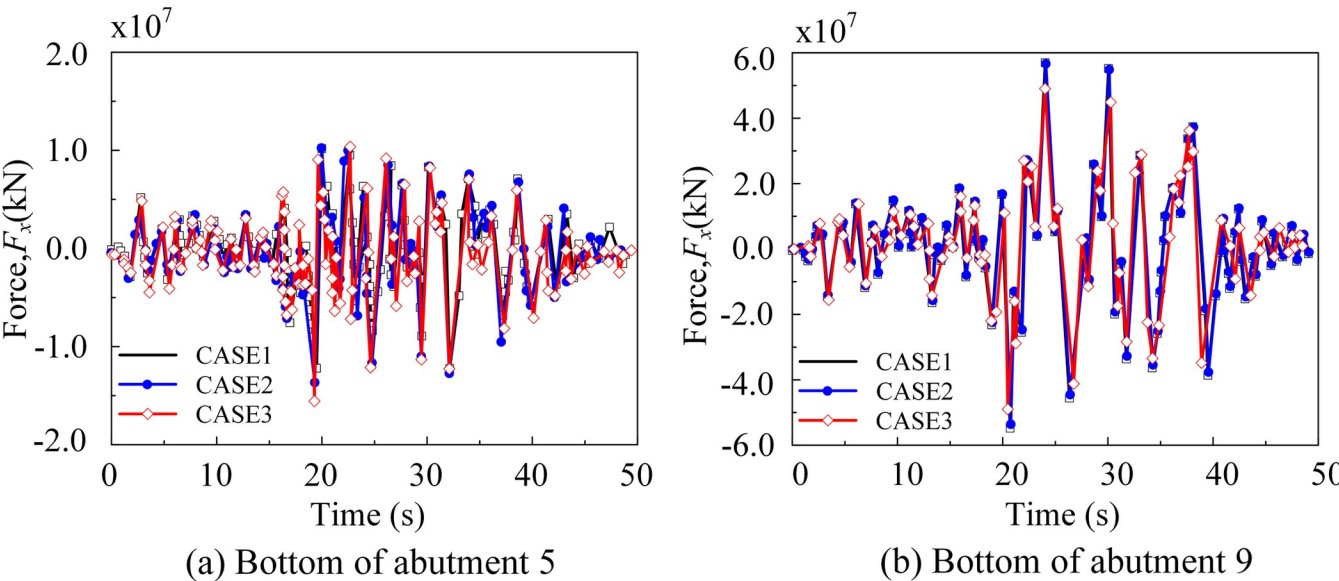

**Fig 32. Time history of shear force at the bottom of the abutment (*x*-direction).** (a) Time history of shear force at the bottom of abutment 5 (*x*-direction). (b) Time history of shear force at the bottom of abutment 9 (*x*-direction).

transformation, the free-field when a plane wave was incident in any direction was obtained. This approach incorporates the concentrated mass explicit finite element method to achieve numerical wave analysis of plane wave scattering problems with incidence from any spatial direction. Through numerical examples, the effectiveness of this method was validated, highlighting the significant influence of the incident angle on seismic response. Through numerical computations, the following conclusions have been derived:

1. This paper utilizes the MPI protocol and FORTRAN language to implement a parallel scheme for the dynamic analysis of three-dimensional time-domain SSI under spatial seismic wave incidence.

2. Under the influence of obliquely incident seismic waves, structures exhibit varying patterns in acceleration response corresponding to changes in the angle of incidence. Under SV wave incidents, when the incident angle $\theta_2 = 0°$, a comparative analysis reveals no time difference in the displacement response. However, when the second spatial incidence angle $\theta_2 \neq 0°$, the vibration onset in all three directions and the arrival time of peak displacement response were delayed, consistent with theoretical calculations.

3. As the seismic wave incidence angle increases, the response of the near-source side abutment increases while that of the far-source side abutment decreases. The displacement response of the main girder in the approach bridge decreases with the oblique incidence angle, particularly when the incidence angle is perpendicular to the valley slope, resulting in the same peak displacement response for the main girder in the approach bridge near the source. The observed responses of the bridge under seismic wave action conform to the expected patterns, thus validating the accuracy of the proposed methodology from a practical perspective.

This paper proposes a computational method under seismic wave input from any direction and validates its feasibility and effectiveness through numerical examples. Considering the complexity of seismic wave propagation at the site, such as non-uniform seismic input, the

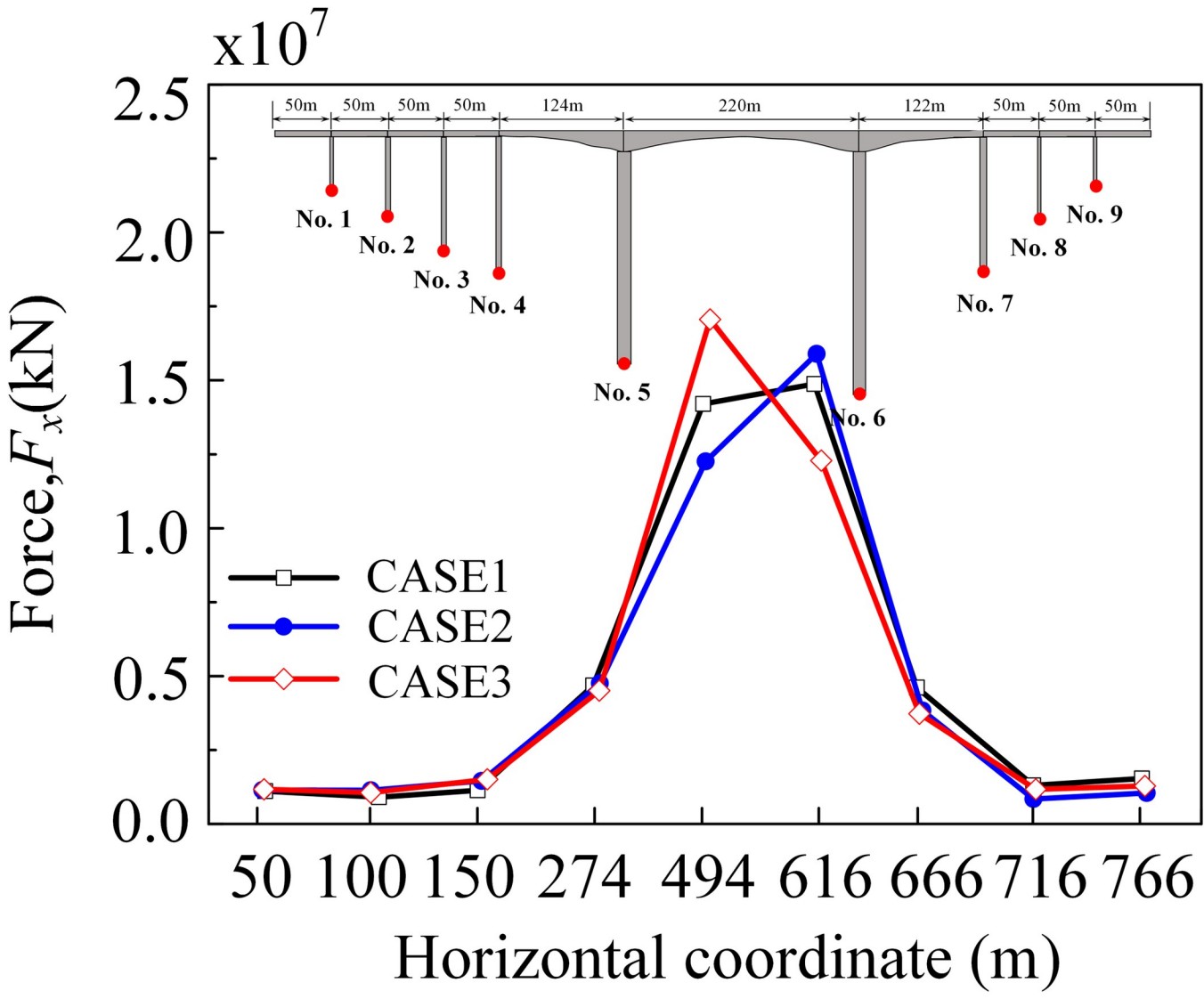

**Fig 33. Peak shear force at the bottom of the abutment (*x*-direction).**

influence of the seismic source, and site nonlinearity, the accuracy of the computed results still requires comparison and verification with engineering experiments and field monitoring data. Therefore, subsequent verification and validation (V&V) are necessary to ensure the accuracy of the computational method.

## Supporting information

**S1 Data. Original data for figures in the article.**
(ZIP)

## Acknowledgments

I would like to express my sincere gratitude to my PhD supervisor, Prof Shaolin Chen, for sharing their methodology and experience with me and making this work possible. I sincerely thank my postdoctoral co-supervisor, Prof Qingbiao Wang, for all the help he provided.

## Author Contributions

**Conceptualization:** Hao Lv.

**Data curation:** Hao Lv.

**Formal analysis:** Hao Lv.

**Funding acquisition:** Hao Lv.

**Methodology:** Hao Lv.

**Validation:** Hao Lv.

**Visualization:** Hao Lv.

**Writing – original draft:** Hao Lv.

**Writing – review & editing:** Hao Lv.

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
