## [Decision Letter · Decision Letter 0]

8 Apr 2024

PONE-D-24-11204Implementation and Application of a Method for Three-dimensional Wave Scattering Problem with Arbitrary Incident Angle of WavesPLOS ONE

Dear Dr. Lv,

Thank you for submitting your manuscript to PLOS ONE. After careful consideration, we feel that it has merit but does not fully meet PLOS ONE’s publication criteria as it currently stands. Therefore, we invite you to submit a revised version of the manuscript that addresses the points raised during the review process.

We look forward to receiving your revised manuscript.

Kind regards,

Dr. S. M. Anas, Ph.D.(Structural Engg.), M.Tech(Earthquake Engg.)

Academic Editor

PLOS ONE

 [I would like to express my sincere gratitude to my PhD supervisor, Prof Shaolin Chen, for sharing their methodology and experience with me and making this work possible. I sincerely thank my postdoctoral co-supervisor, Prof Qingbiao Wang, for providing fund support (the National Natural Science Foundation of China (No. 52278359)). The authors gratefully acknowledge the financial support of this work by the Postdoctoral Fellowship Program of CPSF under Grant Number: GZC20231496.].  

[ I would like to express my sincere gratitude to my PhD supervisor, Prof Shaolin Chen, for sharing their methodology and experience with me and making this work possible. I sincerely thank my postdoctoral co-supervisor, Prof Qingbiao Wang, for providing fund support (the National Natural Science Foundation of China (No. 52278359)). The authors gratefully acknowledge the financial support of this work by the Postdoctoral Fellowship Program of CPSF under Grant Number:GZC20231496]

 [I would like to express my sincere gratitude to my PhD supervisor, Prof Shaolin Chen, for sharing their methodology and experience with me and making this work possible. I sincerely thank my postdoctoral co-supervisor, Prof Qingbiao Wang, for providing fund support (the National Natural Science Foundation of China (No. 52278359)). The authors gratefully acknowledge the financial support of this work by the Postdoctoral Fellowship Program of CPSF under Grant Number: GZC20231496.]

6. In the online submission form, you indicated that [All data, models, or code generated or used during the study are available from the corresponding author by request.]

Additional Editor Comments:

Dear Corresponding Author,

I hope this email finds you well.

I am writing to inform you about the decision regarding your manuscript titled "Implementation and Application of a Method for Three-dimensional Wave Scattering Problem with Arbitrary Incident Angle of Waves" (PONE-D-24-11204), which you submitted to PLOS ONE.

After careful consideration and review by two independent peer reviewers, I have reached a decision. One reviewer suggested minor revisions, while the other recommended major revisions. Following their recommendations and my preliminary assessment of the manuscript, I have decided to request major revisions. Please note that this decision is subject to the final approval of the editorial board.

I would like to emphasize the importance of addressing the concerns and suggestions raised by both reviewers thoroughly. Your revisions should aim to improve the clarity, coherence, and quality of the manuscript to meet the standards of PLOS ONE.

I have attached the reviewers' comments and suggestions for your reference. Please take these into account while revising your manuscript. Additionally, I encourage you to provide a detailed response to each comment, explaining how you have addressed or incorporated the feedback into the revised manuscript.

Once you have completed the revisions, please submit the revised manuscript along with a point-by-point response to the reviewers' comments through the PLOS ONE submission system. Kindly ensure that you highlight all changes made to the manuscript to facilitate the review process.

Important note from Academic Editor, Dr. S. M. Anas:

I wish to highlight that it is not obligatory to cite the papers recommended by the reviewers in your revised manuscript. The decision to include these suggested papers is entirely yours. While the reviewers provided these recommendations to enhance the quality and credibility of your research, you have the freedom to exclude any suggested papers if you believe they are not relevant or do not contribute value to your study.

Should you have any questions or need further clarification, please do not hesitate to contact me. I am here to assist you throughout the revision process.

Thank you for your submission to PLOS ONE, and I look forward to receiving your revised manuscript.

Best regards,

Dr. S. M. Anas

Academic Editor

PLOS ONE

Reviewers' comments:

Reviewer's Responses to Questions

**Comments to the Author**

1. Is the manuscript technically sound, and do the data support the conclusions?

Reviewer #1: Yes

Reviewer #2: Yes

2. Has the statistical analysis been performed appropriately and rigorously? 

Reviewer #1: Yes

Reviewer #2: Yes

3. Have the authors made all data underlying the findings in their manuscript fully available?

Reviewer #1: Yes

Reviewer #2: Yes

4. Is the manuscript presented in an intelligible fashion and written in standard English?

Reviewer #1: Yes

Reviewer #2: Yes

5. Review Comments to the Author

Reviewer #1: In this paper, a novel method was proposed for inputting plane seismic wave incidents. Initially, the two-dimensional free-field problem was analyzed using the transfer matrix method. Then, a three-dimensional free-field motion was calculated by transforming between two coordinate systems. However, there are several issues that must be addressed by the authors before the paper is to be published. These comments are listed as follows:

1- It is recommended to express the sentences of the paper in the third person. Presenting these items is not particularly appropriate in the abstract section.

2- The abstract could be written better. The innovation of the study as well as the step-by-step research process should be mentioned in it.

3- As can be seen (Section 2.1, lines 143 to 146), the approach used in the paper is a numerical method. If so, it is recommended to mention this issue in the title of the manuscript. For example, the title can be modified as follows: “A numerical approach applied to three-dimensional wave scattering problems subjected to obliquely propagating incident waves”.

4- In section 2, line 130, it is mentioned that an analytical response is used! It is mandatory to mention the reference.

5- The author has well pointed out the capability and efficiency of the used approach. In order to complete the paper, it is recommended to mention some limitations of the method used at the end of the conclusion section.

6- The number of paragraphs in the paper is very high. It is better to reduce its number and edit it.

7- The section of the conclusion is written very poorly. The authors should revise this section.

8- Some references in wave propagation problems by other numerical methods (such as the boundary element method) are missed. Here are some papers the authors should have known:

https://link.springer.com/article/10.1007/s11803-020-0603-9

https://link.springer.com/article/10.1007/s41062-019-0257-5

https://www.equsci.org.cn/en/article/doi/10.29382/eqs-2018-0137-3

https://www.tandfonline.com/doi/abs/10.1080/13632469.2021.1927886

9- The paper requires extensive editorial work.

Reviewer #2: Summary: Accurate modeling of artificial boundary conditions and wave inputs is vital for numerical simulations of wave scattering in semi-infinite domains. Traditional methods focus on one or two-dimensional free-field problems, limiting their ability to handle arbitrary incident directions in three-dimensional scenarios. This paper proposes a novel approach for inputting plane seismic waves from any direction. It employs a combination of transfer matrix analysis for two-dimensional free-field problems and transformation between coordinate systems for three-dimensional motion. This method, integrated with the Finite Element Method (FEM), facilitates input for various boundaries. Numerical examples demonstrate its effectiveness in analyzing seismic responses, particularly in nuclear power plants and deep-water bridges, confirming its accuracy in handling complex scenarios.

Observations & Suggestions: For the analysis of three-dimensional fluctuation scattering problems, the conventional approach typically employs free-field analysis in one or two dimensions, often focusing on perpendicular or parallel incidence relative to one of the coordinate planes containing the vertical axis. However, this method restricts the incident directions of fluctuations in three-dimensional space and inadequately captures the true three-dimensional effects of ground vibration input. To address this limitation, it becomes imperative to analyze the free field at arbitrary fluctuation incident directions in three-dimensional space, providing inputs for the comprehensive analysis of fluctuation scattering in three-dimensional problems.

Huang (2015) proposed a solution for homogeneous half-space-free field incidents in any direction by combining the direct incident wave with the reflected wave from the ground surface. Nevertheless, when tackling the free field analysis of horizontally layered sites, this method encounters challenges in tracking the complex reflected and transmitted waves, thereby intensifying the difficulty of solving the problem. To make the kinds of literature sections richer the authors are advised to include the following references:

Barak, M. S., Kumar, M., Kumari, M., & Singh, A. (2020). Inhomogeneous wave propagation in partially saturated soils. Wave Motion, 93, 102470.

Barak, M. S., Kumari, M., & Kumar, M. (2018). Effect of local fluid flow on the propagation of plane waves at an interface of water/double-porosity solid with an underlying uniform elastic solid. Ocean Engineering, 147, 195-205.

Kumari, M., Barak, M. S., & Kumar, M. (2017). Seismic reflection and transmission coefficients of a single layer sandwiched between two dissimilar poroelastic solids. Petroleum Science, 14(4), 676-693.

Recommendations: The work done in this manuscript is appreciable, I recommend the manuscript for possible publication with incorporating the suggestions.

6. PLOS authors have the option to publish the peer review history of their article (what does this mean?). If published, this will include your full peer review and any attached files.

Reviewer #1: **Yes: **Mehdi Panji

Reviewer #2: **Yes: **M.S. Barak

---

## [Author Response · Author response to Decision Letter 0]

29 Apr 2024

Response to Reviewers

Thank you very much for giving us the opportunity to revise the manuscript. We have carefully read the comments and tried our best to modify the manuscript. The revised parts are marked in red. Our response to the comments of the reviewers is as follows.

Review Comments to the Author

Reviewer #1: In this paper, a novel method was proposed for inputting plane seismic wave incidents. Initially, the two-dimensional free-field problem was analyzed using the transfer matrix method. Then, a three-dimensional free-field motion was calculated by transforming between two coordinate systems. However, there are several issues that must be addressed by the authors before the paper is to be published. These comments are listed as follows:

Response to Reviewer #1:

Thank you for your time and patience. We made changes practically exactly as you suggested. In the revised manuscript, we highlight in red font the added text in response to the reviewers’ comments. We hope we addressed all your concerns.

1- It is recommended to express the sentences of the paper in the third person. Presenting these items is not particularly appropriate in the abstract section. 

Response: Thank you for your time and patience. We followed your suggestion and modified and polished the language of the full text. The revised parts are marked in red.

2- The abstract could be written better. The innovation of the study as well as the step-by-step research process should be mentioned in it. 

Response: Thank you. As per your guidance, we have reformulated the abstract section, elucidating the novelty and the incremental research process therein. The revised parts are marked in red.

On line 22 page 2, we rewrote this paragraph as follows:

“Accurately modeling artificial boundary conditions and wave inputs is paramount for numerical simulations of wave scattering in semi-infinite domains within seismic engineering. Traditionally, analysts focused on one- or two-dimensional free-field problems to determine wave inputs, primarily for vertically incident plane waves or obliquely incident waves parallel to two axes. However, these methods were inadequate for handling arbitrary incident directions in three-dimensional scenarios. This paper proposes a method for modeling seismic wave incidents in arbitrary directions. The basic theory of viscoelastic boundaries is leveraged, and a plane containing an arbitrary incident direction and the vertical coordinate axis is selected to establish a two-dimensional plane coordinate system. The two-dimensional free-field problem in this coordinate system is derived using the transfer matrix method. Subsequently, displacement, velocity, and stress are converted into the coordinate system where the three-dimensional calculation model is located, providing input for the three-dimensional scattering problem. Furthermore, the implementation of transmitting boundary conditions and viscoelastic boundary wave inputs is presented to enable incident wave scattering problems at any angle of the plane. The effect of oblique-incidence soil-structure dynamic interaction is also discussed, focusing on the parallel technology method adopted in this paper. With the relatively mature technology route and method, together with nuclear power systems and large-span deep-water bridge models, through examples of comparative analysis, qualitative and quantitative analyses are made on the impact on the soil mass, foundation, and structure when the seismic wave is an oblique incident.”

3- As can be seen (Section 2.1, lines 143 to 146), the approach used in the paper is a numerical method. If so, it is recommended to mention this issue in the title of the manuscript. For example, the title can be modified as follows: “A numerical approach applied to three-dimensional wave scattering problems subjected to obliquely propagating incident waves”. 

Response: Thank you. 

Following your advice, we believe that the title you suggested better encapsulates the core issue of this paper: the implementation of numerical methods. The original title may have leaned more towards the seismic wave incidence problem using arbitrary angle methods, which might not be sufficiently comprehensive. Therefore, we have revised the paper title according to your recommendation.

4- In section 2, line 130, it is mentioned that an analytical response is used! It is mandatory to mention the reference. 

Response: Thanks. As per your suggestion, we have added relevant references here and labelled them in the text.

On line 126 page 6, we add the following reference:

[30] Thomson W T. Transmission of Elastic Waves through a Stratified Solid Medium. Journal of Applied Physics. 1950; 21(2):89-93. https://doi.org/10.1063/1.1699629

[31] Haskell, N. The dispersion of surface waves on multilayer media. Bull.Seism.Soc.AM 1953; 43: 17-34. https://doi.org/10.1785/BSSA0430010017

[33] Liao Z P. Introduction to wave motion theories in engineer. Science Press, 2002. (in chinese).

5- The author has well pointed out the capability and efficiency of the used approach. In order to complete the paper, it is recommended to mention some limitations of the method used at the end of the conclusion section. 

Response: Thanks for your comments. Thank you. We followed your suggestion and modified the end section. The revised parts are marked in red.

On line 686 page 33, we rewrote this paragraph as follows:

“While this paper proposed a computational method under the seismic wave input from any direction and validates its feasibility and effectiveness through numerical examples. Considering the complexity of seismic wave propagation in the site, such as non-uniform seismic input, the influence of the seismic source, and site nonlinearity, the accuracy of the computed results still requires comparison and verification with engineering experiments and field monitoring data. Therefore, subsequent Verification and Validation (V&V) are necessary to ensure the accuracy of the computational method.”

6- The number of paragraphs in the paper is very high. It is better to reduce its number and edit it. 

Response: Thank you for your time and patience. Following your advice, we have merged relevant paragraphs in the paper, reducing the overall number of paragraphs.

7- The section of the conclusion is written very poorly. The authors should revise this section.

Response: Thank you very much for bringing our attention to the structures of “Conclusion”. We followed your suggestion and modified and polished the language of the conclusion. The revised parts are marked in red.

On line 657 page 32, we rewrote this paragraph as follows:

“This paper presents a numerical wave analysis of the scattering of plane waves incident from any spatial direction by combining the transmission boundary and viscoelastic boundary simulations of wave scattering problems. Utilizing the transfer matrix method and coordinate transformation, the free field when a plane wave was incident in any direction was obtained. This approach incorporates the concentrated mass explicit finite element method to achieve numerical wave analysis of plane wave scattering problems with incidence from any spatial direction. Through numerical examples, the effectiveness of this method was validated, highlighting the significant influence of the incident angle on seismic response. Through numerical computations, the following conclusions have been derived:

(1) This paper utilizes the MPI protocol and FORTRAN language to implement a parallel scheme for the dynamic analysis of three-dimensional time-domain SSI under spatial seismic wave incidence.

(2) Under the influence of obliquely incident seismic waves, structures exhibit varying patterns in acceleration response corresponding to changes in the angle of incidence. Under SV wave incidents, when the incident angle θ2 = 0°, a comparative analysis reveals no time difference in the displacement response. However, when the second spatial incidence angle θ2 ≠ 0°, the vibration onset in all three directions and the arrival time of peak displacement response were delayed, consistent with theoretical calculations.

(3) As the seismic wave incidence angle increases, the response of the near-source side abutment increases while that of the far-source side abutment decreases. The displacement response of the main girder in the approach bridge decreases with the oblique incidence angle, particularly when the incidence angle was perpendicular to the valley slope, resulting in the same peak displacement response for the main girder in the approach bridge near the source. The observed responses of the bridge under seismic wave action conform to the expected patterns, thus validating the accuracy of the proposed methodology from a practical perspective.

While this paper proposed a computational method under the seismic wave input from any direction and validates its feasibility and effectiveness through numerical examples. Considering the complexity of seismic wave propagation in the site, such as non-uniform seismic input, the influence of the seismic source, and site nonlinearity, the accuracy of the computed results still requires comparison and verification with engineering experiments and field monitoring data. Therefore, subsequent Verification and Validation (V&V) are necessary to ensure the accuracy of the computational method.”

8- Some references in wave propagation problems by other numerical methods (such as the boundary element method) are missed. Here are some papers the authors should have known:

https://link.springer.com/article/10.1007/s11803-020-0603-9

https://link.springer.com/article/10.1007/s41062-019-0257-5

https://www.equsci.org.cn/en/article/doi/10.29382/eqs-2018-0137-3

https://www.tandfonline.com/doi/abs/10.1080/13632469.2021.1927886

https://link.springer.com/article/10.1007/s11803-020-0603-9

https://link.springer.com/article/10.1007/s41062-019-0257-5

https://www.equsci.org.cn/en/article/doi/10.29382/eqs-2018-0137-3

https://www.tandfonline.com/doi/abs/10.1080/13632469.2021.1927886

Response: Thank you. I apologize for the inconvenience, but I need to further strengthen my understanding of the outcomes related to using numerical methods to solve wave propagation problems. Following your suggestion, I have incorporated citations to the relevant literature at the appropriate places in the text (References [13] and [17]). Thank you for your recommendation.

[13] Panji, M., Ansari, B. Anti-plane seismic ground motion above twin horseshoe-shaped lined tunnels. Innov. Infrastruct. Solut. 2020; 5: 7. https://doi.org/10.1007/s41062-019-0257-5

[17] Panji, M., Habibivand, M. Seismic analysis of semi-sine shaped alluvial hills above subsurface circular cavity. Earthq. Eng. Eng. Vib. 2020; 19: 903-917. https://doi.org/10.1007/s11803-020-0603 -9

9- The paper requires extensive editorial work. 

Response: Thank you for your time and patience. We followed your suggestion and modified and polished the language of the full text. The revised parts are marked in red.

Reviewer #2: 

Summary: Accurate modeling of artificial boundary conditions and wave inputs is vital for numerical simulations of wave scattering in semi-infinite domains. Traditional methods focus on one or two-dimensional free-field problems, limiting their ability to handle arbitrary incident directions in three-dimensional scenarios. This paper proposes a novel approach for inputting plane seismic waves from any direction. It employs a combination of transfer matrix analysis for two-dimensional free-field problems and transformation between coordinate systems for three-dimensional motion. This method, integrated with the Finite Element Method (FEM), facilitates input for various boundaries. Numerical examples demonstrate its effectiveness in analyzing seismic responses, particularly in nuclear power plants and deep-water bridges, confirming its accuracy in handling complex scenarios. 

Observations & Suggestions: For the analysis of three-dimensional fluctuation scattering problems, the conventional approach typically employs free-field analysis in one or two dimensions, often focusing on perpendicular or parallel incidence relative to one of the coordinate planes containing the vertical axis. However, this method restricts the incident directions of fluctuations in three-dimensional space and inadequately captures the true three-dimensional effects of ground vibration input. To address this limitation, it becomes imperative to analyze the free field at arbitrary fluctuation incident directions in three-dimensional space, providing inputs for the comprehensive analysis of fluctuation scattering in three-dimensional problems. 

Huang (2015) proposed a solution for homogeneous half-space-free field incidents in any direction by combining the direct incident wave with the reflected wave from the ground surface. Nevertheless, when tackling the free field analysis of horizontally layered sites, this method encounters challenges in tracking the complex reflected and transmitted waves, thereby intensifying the difficulty of solving the problem. To make the kinds of literature sections richer the authors are advised to include the following references: 

Barak, M. S., Kumar, M., Kumari, M., & Singh, A. (2020). Inhomogeneous wave propagation in partially saturated soils. Wave Motion, 93, 102470.

Barak, M. S., Kumari, M., & Kumar, M. (2018). Effect of local fluid flow on the propagation of plane waves at an interface of water/double-porosity solid with an underlying uniform elastic solid. Ocean Engineering, 147, 195-205.

Kumari, M., Barak, M. S., & Kumar, M. (2017). Seismic reflection and transmission coefficients of a single layer sandwiched between two dissimilar poroelastic solids. Petroleum Science, 14(4), 676-693.

Recommendations: The work done in this manuscript is appreciable, I recommend the manuscript for possible publication with incorporating the suggestions. 

Response to Reviewer #2:

Response: Thank you for carefully reading our manuscript and for your suggestions on how to improve it. I apologize for the inconvenience, but I need to further strengthen my understanding of the outcomes related to using numerical methods to solve wave propagation problems. Following your suggestion, I have incorporated citations to the relevant literature at the appropriate places in the text (References [27] and [28]). Thank you for your recommendation.

[27] Kumari, M., Barak, M. S., Kumar, M. Seismic reflection, and transmission coefficients of a single layer sandwiched between two dissimilar poroelastic solids. Petroleum Science. 2017; 14(4): 676-693. https://doi.org/10.1007/s12182-017-0195-9

[28] Barak, M. S., Kumar, M., Kumari, M., Singh, A. Inhomogeneous wave propagation in partially saturated soils. Wave Motion.2020;93:102470. https://doi.org/10.1016/j.wavemoti.2019. 102470

---

## [Decision Letter · Decision Letter 1]

8 May 2024

PONE-D-24-11204R1A numerical approach applied to three-dimensional wave scattering problems subjected to obliquely propagating incident wavesPLOS ONE

Dear Dr. Lv,

Thank you for submitting your manuscript to PLOS ONE. After careful consideration, we feel that it has merit but does not fully meet PLOS ONE’s publication criteria as it currently stands. Therefore, we invite you to submit a revised version of the manuscript that addresses the points raised during the review process.

We look forward to receiving your revised manuscript.

Kind regards,

Dr. S. M. Anas, Ph.D.(Structural Engg.), M.Tech(Earthquake Engg.)

Academic Editor

PLOS ONE

Journal Requirements:

Additional Editor Comments:

Dear Authors,

I hope this email finds you well.

I am writing to inform you about the status of your revised manuscript entitled "A numerical approach applied to three-dimensional wave scattering problems subjected to obliquely propagating incident waves" [PONE-D-24-11204R1]. The revised version was sent to the previous reviewers for reevaluation, and I'm pleased to inform you that both reviewers recommended the paper for publication and expressed satisfaction with your responses to their comments.

However, upon my preliminary assessment of the revised manuscript, I observed that you have added references suggested by the reviewers without providing a relevant reason for their inclusion. Therefore, I have decided to request a minor revision. I kindly ask you to review the suggested references and determine whether they are truly relevant to your study. If they are not relevant, please remove them. If they are relevant, kindly provide a valid reason for their addition.

Please address this revision request at your earliest convenience and resubmit the manuscript accordingly. Feel free to reach out if you have any questions or need further clarification.

Thank you for your attention to this matter.

Best regards,

Dr. S. M. Anas

Academic Editor

PLOS ONE

Reviewers' comments:

Reviewer's Responses to Questions

**Comments to the Author**

1. If the authors have adequately addressed your comments raised in a previous round of review and you feel that this manuscript is now acceptable for publication, you may indicate that here to bypass the “Comments to the Author” section, enter your conflict of interest statement in the “Confidential to Editor” section, and submit your "Accept" recommendation.

Reviewer #1: All comments have been addressed

Reviewer #2: All comments have been addressed

2. Is the manuscript technically sound, and do the data support the conclusions?

Reviewer #1: Yes

Reviewer #2: Yes

3. Has the statistical analysis been performed appropriately and rigorously? 

Reviewer #1: Yes

Reviewer #2: Yes

4. Have the authors made all data underlying the findings in their manuscript fully available?

Reviewer #1: Yes

Reviewer #2: No

5. Is the manuscript presented in an intelligible fashion and written in standard English?

Reviewer #1: Yes

Reviewer #2: Yes

6. Review Comments to the Author

Reviewer #1: All comments have been addressed by the author as well. The manuscript can be accepted in the present form.

Reviewer #2: I appreciate the point wise reply of the authors to each comments and suggessions made by the reviewers, and I recommend the manuscript for possible publication in the current form.

7. PLOS authors have the option to publish the peer review history of their article (what does this mean?). If published, this will include your full peer review and any attached files.

Reviewer #1: **Yes: **Mehdi Panji

Reviewer #2: **Yes: **M.S.Barak

---

## [Author Response · Author response to Decision Letter 1]

13 May 2024

Response to Reviewers

Thank you very much for giving us the opportunity to revise the manuscript. We have carefully read the comments and modified the manuscript. The revised parts are marked in red. Our response to the comments of the reviewers is as follows: 

Response to Reviewer:

1. Please include a figure label and title for figures to 1 to 33 in your main manuscript.

On line 694, page 33, we have added figure labels and titles as follows:

S1 Fig. Schematic diagram of soil structure interaction model under spatial incidence of seismic wave.

S2 Fig. Geometrical relation of the multi-transmitting boundary.

S3 Fig. Schematic diagram of the finite calculation area and equivalent spring-damper system on the artificial boundary.

S4 Fig. Equal force and stress direction of interfaces in all directions.

S5 Fig. Finite element model of the site.

S6 Fig. Input wave.

S7 Fig. Displacement of half space subjected to SV wave oblique incidence.

S8 Fig. The horizontal displacement time history of reference point A under SV wave oblique incidence.

S9 Fig. An example model schematic diagram.

S10 Fig. Input plus wave.

S11 Fig. Displacement in y-direction at observation points. Note: T and V represent the transmitting boundary and the viscous-spring artificial boundary.

S12 Fig. Displacement of the observation points (SH wave).

S13 Fig. Displacement of the observation points.

S14 Fig. An example model schematic diagram of the NPP.

S15 Fig. The input wave and the cut-off frequency.

S16 Fig. Schematic diagram of the partitioned analysis of soil-structure interaction and the position of the reference points of the soil subdomain.

S17 Fig. Displacement response of soil reference points A11 and A31.

S18 Fig. Displacement response of soil reference points A13 and A33.

S19 Fig. Displacement response of soil reference points A21 and A23.

S20 Fig. Displacement response of the foundation reference point A22.

S21 Fig. The response of the structure.

S22 Fig. General layout of the bridge.

S23 Fig. Finite element model of the bridge.

S24 Fig. Acceleration response spectrum.

S25 Fig. Ground motion and its Fourier amplitude spectrum.

S26 Fig. Schematic diagram of the calculation site.

S27 Fig. Displacement at the bottom of pier 9 (x-direction).

S28 Fig. Peak displacement response at the bottom of the abutment (x-direction).

S29 Fig. Displacement response of approach bridge (x-direction).

S30 Fig. Peak displacement response at the bearing of the main girder of the approach bridge (x-direction).

S31 Fig. Displacement response of approach bridge (x-direction).

S32 Fig. Time history of shear force at the bottom of the abutment (x-direction).

S33 Fig. Peak shear force at the bottom of the abutment (x-direction).

S1 Data. (Zip) Original data for figures in the article.

Additionally, we have renamed the images and data in the supplementary files Figures.zip and S1 Data.zip to comply with the journal's requirements.

We appreciate your review and valuable suggestions once again. Should you have any further recommendations or requests regarding our modifications, please feel free to inform us.

---

## [Editor Report · Decision Letter 2]

15 May 2024

PONE-D-24-11204R2A numerical approach applied to three-dimensional wave scattering problems subjected to obliquely propagating incident wavesPLOS ONE

Dear Dr. Lv,

Thank you for submitting your manuscript to PLOS ONE. After careful consideration, we feel that it has merit but does not fully meet PLOS ONE’s publication criteria as it currently stands. Therefore, we invite you to submit a revised version of the manuscript that addresses the points raised during the review process.

We look forward to receiving your revised manuscript.

Kind regards,

Dr. S. M. Anas, Ph.D.(Structural Engg.), M.Tech(Earthquake Engg.)

Academic Editor

PLOS ONE

Journal Requirements:

Additional Editor Comments:

**Dear Authors**,

**I hope this email finds you well.**

**I am writing to inform you about the status of your second revised manuscript titled "A numerical approach applied to three-dimensional wave scattering problems subjected to obliquely propagating incident waves" [PONE-D-24-11204R2]. Upon review of the second revision, it has come to our attention that the minor revision requested in the previous communication has not been adequately addressed in your author response form.**

**In our previous correspondence, I had requested a minor revision concerning the inclusion of references suggested by the reviewers without providing a relevant reason for their addition. Although your second revised manuscript has been submitted, the concerns outlined in the previous revision request remain unaddressed.**

**Therefore, I kindly ask you to carefully review the comments provided by the editor and ensure that all requested revisions are addressed appropriately in your resubmission. Specifically, please provide valid reasons for the inclusion of any additional references suggested by the reviewers.**

**We understand that revisions can be time-consuming, but it is essential to ensure the quality and relevance of the manuscript before proceeding with the publication process. Your prompt attention to this matter would be greatly appreciated.**

**Should you have any questions or require further clarification regarding the revision process, please do not hesitate to contact us.**

**Thank you for your cooperation and understanding.**

**Best regards,**

**Dr. S. M. Anas**

**Academic Editor**

**PLOS ONE**

---

## [Author Response · Author response to Decision Letter 2]

16 May 2024

Dear Dr. S. M. Anas,

Academic Editor

PLOS ONE

I apologize for taking up so much of your time. Thank you very much for carefully reviewing our manuscript, providing positive feedback, and offering suggestions for revision.

I would like to clarify that regarding the issues raised in the second revised manuscript titled "A numerical approach applied to three-dimensional wave scattering problems subjected to obliquely propagating incident waves" [PONE-D-24-11204R1], we have made modifications and provided explanations in our response to the first round of minor revisions. The revised manuscript was completed on May 11th and returned as requested. Subsequently, on May 13th, we received the request to add figure labels and captions for Figures 1 to 33. We have made the necessary changes and returned the manuscript accordingly. In this revision, we replaced the response to the May 11th modifications (regarding the addition or removal of references) with the response addressing the modification of "figure labels and captions," which may have caused you to overlook our relevant statements and feedback on your suggestions. We sincerely apologize for this oversight. Therefore, in this response, we will combine the feedback from the last two rounds of revisions and provide a unified reply.

Thank you again for your evaluation of our revised manuscript and for providing valuable feedback.

Best regards,

Dr. Hao Lv

College of Safety and Environmental Engineering, Shandong University of Science and Technology, Qingdao 266590, China

The relevant modification responses are as follows：

Response to Reviewers

Thank you for your evaluation of our revised manuscript and for providing valuable feedback. In response to your guidance, we have carefully reviewed the suggested references and reassessed their relevance to our research.

During the review process, we indeed noted the issue you mentioned regarding the addition of suggested references without providing relevant justifications, which was an inappropriate practice. Consequently, we have conducted a comprehensive review of the literature and confirmed its alignment with the focus of our study.

We acknowledge the importance of ensuring the relevance of references to the research topic in enhancing the quality of the manuscript. Hence, we have removed references that are not pertinent to our study. The revised parts are marked in red. 

Response to Reviewer（first minor revised manuscript，May 11th）:

Additional Editor Comments:

Dear Authors,

I hope this email finds you well.

I am writing to inform you about the status of your revised manuscript entitled "A numerical approach applied to three-dimensional wave scattering problems subjected to obliquely propagating incident waves" [PONE-D-24-11204R1]. The revised version was sent to the previous reviewers for reevaluation, and I'm pleased to inform you that both reviewers recommended the paper for publication and expressed satisfaction with your responses to their comments.

However, upon my preliminary assessment of the revised manuscript, I observed that you have added references suggested by the reviewers without providing a relevant reason for their inclusion. Therefore, I have decided to request a minor revision. I kindly ask you to review the suggested references and determine whether they are truly relevant to your study. If they are not relevant, please remove them. If they are relevant, kindly provide a valid reason for their addition.

Please address this revision request at your earliest convenience and resubmit the manuscript accordingly. Feel free to reach out if you have any questions or need further clarification.

Thank you for your attention to this matter.

Best regards,

Dr. S. M. Anas

Academic Editor

PLOS ONE

And the third minor revised manuscript，May 15th

Additional Editor Comments:

Dear Authors,

I hope this email finds you well.

I am writing to inform you about the status of your second revised manuscript titled "A numerical approach applied to three-dimensional wavescattering problems subjected to obliquely propagating incident waves" [PONE-D-24-11204R2]. Upon review of the second revision, it has come toour attention that the minor revision requested in the previous communication has not been adequately addressed in your author response form.

In our previous correspondence, I had requested a minor revision concerning the inclusion of references suggested by the reviewers withoutproviding a relevant reason for their addition. Although your second revised manuscript has been submitted, the concerns outlined in the previousrevision request remain unaddressed.

Therefore, I kindly ask you to carefully review the comments provided by the editor and ensure that all requested revisions are addressedappropriately in your resubmission. Specifically, please provide valid reasons for the inclusion of any additional references suggested by thereviewers.

We understand that revisions can be time-consuming, but it is essential to ensure the quality and relevance of the manuscript before proceedingwith the publication process. Your prompt attention to this matter would be greatly appreciated.

Should you have any questions or require further clarification regarding the revision process, please do not hesitate to contact us.

Thank you for your cooperation and understanding.

Best regards,

Dr. S. M. Anas

Academic Editor

PLOS ONE

Our response to the comments of the reviewers is as follows:

The number of lines and pages indicated here are those in reference file PONE-D-24-11204_R1.pdf

1- On line 56, page 3,

Based on your suggestion, we have re-evaluated the relevant content of the paper, and have decided to remove reference [13]. The primary rationale behind this decision is the document's emphasis on engineering practice, which is not closely related to the theoretical research on site response analysis in this section. After careful consideration, we have opted to exclude reference [13].

Reference:

[13] Panji, M., Ansari, B. Anti-plane seismic ground motion above twin horseshoe-shaped lined tunnels. Innov. Infrastruct. Solut. 2020; 5: 7. https://doi.org/10.1007/s41062-019-0257-5

2- On line 62, page 3,

In response to the reviewer's suggestions, reference [17] has been incorporated. This decision was driven by the referenced study's utilization of the half-plane time domain boundary element method (BEM) to conduct seismic analysis on semi-sine shaped alluvial hills above circular underground cavities. The study elucidated the significant impact of the hill's material properties and its heterogeneity with the underlying half-space on surface response. This falls within the realm of site analysis methodologies, aligning with the thematic content of this section and enriching the scope of theoretical exploration. The authors deemed its inclusion appropriate.

Reference:

[17] Panji, M., Habibivand, M. Seismic analysis of semi-sine shaped alluvial hills above subsurface circular cavity. Earthq. Eng. Eng. Vib. 2020; 19: 903-917. https://doi.org/10.1007/s11803-020-0603 -9

3- On line 85, page 4,

References [27,28] primarily explore the seismic reflection and transmission characteristics between two dissimilar poroelastic solids, as well as the propagation of inhomogeneous plane harmonic waves in dissipatively partially saturated soils. The research focuses on solving and analyzing the response of saturated soil sites. While this provides valuable insights for extending our research focus to saturated soil sites, it has limited relevance to the theoretical analysis of site response in this section. Therefore, we have decided to cautiously remove references [27,28].

Reference:

[27] Kumari, M., Barak, M. S., Kumar, M. Seismic reflection, and transmission coefficients of a single layer sandwiched between two dissimilar poroelastic solids. Petroleum Science. 2017; 14(4): 676-693. https://doi.org/10.1007/s12182-017-0195-9

[28] Barak, M. S., Kumar, M., Kumari, M., Singh, A. Inhomogeneous wave propagation in partially saturated soils. Wave Motion.2020;93:102470. https://doi.org/10.1016/j.wavemoti.2019. 102470

4- On line 126, page 6, we add the following reference:

[30] Thomson W T. Transmission of Elastic Waves through a Stratified Solid Medium. Journal of Applied Physics. 1950; 21(2):89-93. https://doi.org/10.1063/1.1699629

[31] Haskell, N. The dispersion of surface waves on multilayer media. Bull.Seism.Soc.AM 1953; 43: 17-34. https://doi.org/10.1785/BSSA0430010017

[33] Liao Z. P. Introduction to wave motion theories in engineer. Science Press, 2002. (in chinese).

Following the recommendations of the reviewers and after careful consideration, references [30, 31], and [33] are included in this section. Professor Thomson and Professor Haskell are acknowledged pioneers in the field of engineering theory. The referenced studies in these citations have addressed the propagation issues of waves in site conditions and have provided analytical solutions, thus possessing significant insights and reference value. Therefore, references [30] and [31] have been incorporated at this juncture.

Academician Liao's reference [33] focuses predominantly on discussing practical artificial boundary conditions for simulating infinite domains, thereby systematically expounding on numerical simulation techniques for near-field wave propagation. This technique finds application not only in addressing solid dynamics problems in civil engineering, mechanical engineering, and geophysics but also holds utility for tackling linear and nonlinear dynamics issues in large-scale complex systems and the near-field challenges associated with elastic wave scattering. Moreover, it offers reference value for investigating near-field wave propagation problems in technology domains such as electromagnetics and acoustics. Additionally, the author's research group maintains a fruitful collaboration with Academician Liao, who has provided valuable insights and suggestions for the author's relevant research endeavors.

Following the revision of the relevant literature content, the corresponding references have been renumbered and appropriately modified.

Response to Reviewer（second minor revised manuscript，May 13th）:

1. Please include a figure label and title for Figures to 1 to 33 in your main manuscript.

In the “Supporting Information” section of the article, we have added figure labels and titles as follows:

S1 Fig. Schematic diagram of soil structure interaction model under spatial incidence of seismic wave.

S2 Fig. Geometrical relation of the multi-transmitting boundary.

S3 Fig. Schematic diagram of the finite calculation area and equivalent spring-damper system on the artificial boundary.

S4 Fig. Equal force and stress direction of interfaces in all directions.

S5 Fig. Finite element model of the site.

S6 Fig. Input wave.

S7 Fig. Displacement of half space subjected to SV wave oblique incidence.

S8 Fig. The horizontal displacement time history of reference point A under SV wave oblique incidence.

S9 Fig. An example model schematic diagram.

S10 Fig. Input plus wave.

S11 Fig. Displacement in y-direction at observation points. Note: T and V represent the transmitting boundary and the viscous-spring artificial boundary.

S12 Fig. Displacement of the observation points (SH wave).

S13 Fig. Displacement of the observation points.

S14 Fig. An example model schematic diagram of the NPP.

S15 Fig. The input wave and the cut-off frequency.

S16 Fig. Schematic diagram of the partitioned analysis of soil-structure interaction and the position of the reference points of the soil subdomain.

S17 Fig. Displacement response of soil reference points A11 and A31.

S18 Fig. Displacement response of soil reference points A13 and A33.

S19 Fig. Displacement response of soil reference points A21 and A23.

S20 Fig. Displacement response of the foundation reference point A22.

S21 Fig. The response of the structure.

S22 Fig. General layout of the bridge.

S23 Fig. Finite element model of the bridge.

S24 Fig. Acceleration response spectrum.

S25 Fig. Ground motion and its Fourier amplitude spectrum.

S26 Fig. Schematic diagram of the calculation site.

S27 Fig. Displacement at the bottom of pier 9 (x-direction).

S28 Fig. Peak displacement response at the bottom of the abutment (x-direction).

S29 Fig. Displacement response of approach bridge (x-direction).

S30 Fig. Peak displacement response at the bearing of the main girder of the approach bridge (x-direction).

S31 Fig. Displacement response of approach bridge (x-direction).

S32 Fig. Time history of shear force at the bottom of the abutment (x-direction).

S33 Fig. Peak shear force at the bottom of the abutment (x-direction).

S1 Data. (Zip) Original data for figures in the article.

Additionally, we have renamed the images and data in the supplementary files Figures.zip and S1 Data.zip to comply with the journal's requirements.

We appreciate your review and valuable suggestions once again. Should you have any further recommendations or requests regarding our modifications, please feel free to inform us.

---

## [Editor Report · Decision Letter 3]

17 May 2024

A numerical approach applied to three-dimensional wave scattering problems subjected to obliquely propagating incident waves

PONE-D-24-11204R3

Dear Dr. Lv,

We’re pleased to inform you that your manuscript has been judged scientifically suitable for publication and will be formally accepted for publication once it meets all outstanding technical requirements.

Kind regards,

Dr. S. M. Anas, Ph.D.(Structural Engg.), M.Tech(Earthquake Engg.)

Academic Editor

PLOS ONE

Additional Editor Comments (optional):

Dear Authors,

I hope this email finds you well.

I am writing to inform you that I have thoroughly reviewed the revised manuscript entitled "A numerical approach applied to three-dimensional wave scattering problems subjected to obliquely propagating incident waves" [PONE-D-24-11204R3], which you submitted to PLOS ONE.

After carefully considering the revisions made by you and your co-authors in response to the comments raised earlier, I am pleased to inform you that the manuscript is now suitable for publication. The revisions have addressed the concerns effectively, and the paper maintains its scientific rigor and clarity.

Therefore, I recommend the revised manuscript for publication in PLOS ONE, subject to the approval of the editorial board.

Thank you for your diligence and attention to detail throughout the revision process. Should you have any further questions or concerns, please do not hesitate to contact me.

Warm regards,

Dr. S. M. Anas

Academic Editor

PLOS ONE
---

## [Editor Report · Acceptance letter]

29 May 2024

PONE-D-24-11204R3 

PLOS ONE

Dear Dr. Lv, 

I'm pleased to inform you that your manuscript has been deemed suitable for publication in PLOS ONE. Congratulations! Your manuscript is now being handed over to our production team.

Kind regards, 

on behalf of

Dr. S. M. Anas 

Academic Editor

PLOS ONE